# Sex-specific KDM6A-HNF4A-CREBH network controls lipoprotein cholesterol metabolism and atherosclerosis via epigenetic reprograming of hepatocytes

Lin Chen[1,11], Zhanfang Kang [2,3,4,11], Jennifer Härdfeldt [1,5,6], Ziyi Li[1], Matteo Pedrelli [1,5,6], Qi Li[1], Ruining Lyu[1,7], Philipp Valina Allo[1], Taras Sych[8], Xiangru Zheng[1,9], Peibin Lin[3,4], Jianwen Zeng[2,4], Zhiqiang Huang [1,7], Oihane Garcia-Irigoyen[1], Sviatlana Sukhanava[1], Paolo Parini[1,5,6], Amélie Bonnefond [10], Erdinc Sezgin [8], Bo Angelin [1,5,6], Eckardt Treuter [1] & Rongrong Fan [1,5] ✉

The liver is a central organ controlling lipid and cholesterol metabolism and plays a key role in regulating lipoprotein profiles and cardiovascular disease risk. Males and females show clear differences in cholesterol handling and susceptibility to atherosclerosis, but the molecular basis for these sex-specific effects remains incompletely understood. Here we show that the X-linked histone demethylase 6 A (KDM6A) is essential for maintaining healthy cholesterol metabolism in the liver. Reducing KDM6A levels in human liver cells from females but not males disrupts gene programs involved in lipoprotein regulation linked to cardiovascular disorders. Consistently, female mice lacking KDM6A specifically in hepatocytes develops pro-atherogenic blood lipoprotein profiles and increased atherosclerosis under genetic and dietary stress, whereas males are largely unaffected. Mechanistically, KDM6A cooperates with Hepatocyte Nuclear Factor 4 Alpha (HNF4A) to promote chromatin activation and enable CREBH (encoded by CREB3L3)-dependent transcription of lipid metabolic genes. These findings identify KDM6A as a sex-linked regulator of hepatic cholesterol metabolism.

The liver plays a key role in cholesterol metabolism. It is the major organ to facilitate cholesterol uptake, biosynthesis, excretion and clearance[1]. These processes are governed by coordinated transcriptional networks, epigenetically regulated by transcription factors and coregulators[2,3]. Disturbances of such transcription networks may lead to imbalanced intrahepatic and circulating cholesterol levels and are closely related to human atherosclerotic cardiovascular disease (ACVD)[4–6]. Progress in the past few decades has revealed key

[1]Department of Medicine Huddinge, Karolinska Institutet, Stockholm, Sweden. [2]Guangdong Engineering Technology Research Center of Urinary Continence and Reproductive Medicine, the Affiliated Qingyuan Hospital (Qingyuan People's Hospital), Guangzhou Medical University, Qingyuan, China. [3]Department of Basic Medical Research, the Affiliated Qingyuan Hospital (Qingyuan People's Hospital), Guangzhou Medical University, Qingyuan, China. [4]Department of Urology, the Affiliated Qingyuan Hospital (Qingyuan People's Hospital), Guangzhou Medical University, Qingyuan, China. [5]Cardio Metabolic Unit, Department of Medicine Huddinge, and Department of Laboratory Medicine, Karolinska Institutet, Stockholm, Sweden. [6]Medical Unit of Endocrinology, Theme Inflammation and Ageing, Karolinska University Hospital Huddinge, Stockholm, Sweden. [7]Medical School, Nanjing University, Nanjing, China. [8]Science for Life Laboratory, Department of Women's and Children's Health, Karolinska Institutet, Stockholm, Sweden. [9]Department of Hepatobiliary and Pancreatic Surgery, The Third Affiliated Hospital of Chongqing Medical University, Chongqing, China. [10]University of Lille, INSERM U1283, CNRS UMR 8199, Institut Pasteur de Lille, Lille University Hospital, Lille, France. [11]These authors contributed equally: Lin Chen, Zhanfang Kang. ✉e-mail: rongrong.fan@ki.se

transcription factors (TFs) involved in cholesterol and lipoprotein metabolism in the liver, exemplified by oxysterol-sensing Liver X Receptors (LXRs)[7] and bile acid-sensing Farnesoid X Receptor (FXR)[8,9], both controlling reverse cholesterol transport, bile acid synthesis and excretion via activating the transcription of a group of cholesterol metabolic genes in the hepatocytes. Other well-studied TFs include fatty acid responsive Peroxisome Proliferator-Activated Receptors (PPARs)[10,11] and Sterol Regulatory Element-Binding Protein (SREBPs)[12], some of which are targets of therapeutic agents.

In contrast to TFs, the pathophysiological roles of coregulators in liver cholesterol metabolism remain largely unknown. Recent multi-omics approaches have revealed an intrinsic connection between transcriptome and epigenome which is directly linked to coregulators, some of which have enzymatic activities and regulate histone or DNA modifications[13–15]. Coregulators interplay with TFs and control their transcriptional activity and gene-selectivity by altering the epigenetic chromatin landscape and changing their affinity to TFs and transcriptional activation complexes, amongst many other mechanisms[16–18]. As coregulators interact with multiple TFs, even subtle changes in their expression may lead to a wide spectrum of physiological outcomes associated with various diseases[19,20]. Of note, a few functionally very important epigenetic modulators are located in the sex chromosomes[21]. Some of them can escape X chromosome inactivation and thus become higher expressed in female than in male tissues[21,22].

Lysine Demethylase 6 A (KDM6A) is among the two key X-linked and X-inactivation 'escapee' epigenetic enzymes (the other being KDM5C)[21,22]. The C-terminus of KDM6A contains an enzymatic Jumonji C (JmjC) domain which is able to catalyze the removal of di- and tri-methylation on H3K27 (H3K27me2/3)[22]. Because H3K27me2 and 3 are histone markers for tightly packed 'heterochromatin', demethylases targeting these two modifications help to transform the chromatin to an active state with more access to transcriptional activation complexes[23]. Moreover, KDM6A physically interacts with other coregulator complexes such as Mixed-Lineage Leukemia protein 3/4 (MLL3/4, also named Lysine N-methyltransferase 2 C/2D or KMT2C/2D) and histone acetyltransferase P300/ CBP, and is involved in enhancer activation via modulating chromatin H3K4 methylation (H3K4me) and H3K27 acetylation (H3K27ac)[22]. Indeed, many regulatory functions of KDM6A in the cells do not require its demethylase activities but are dependent on its interacting TFs and coregulators[22]. Because of its important role in shaping chromatin epigenetic landscapes, a number of studies have been conducted in the past years to decipher the physiological function of KDM6A in multiple tissues. So far, most of the studies have been focused on cancer because KDM6A mutations are frequently observed in human cancer patients[22,24–26]. For example, KDM6A expression was lost in 30% of liver cancer patients in a recent study, and removal of KDM6A promoted liver cancer development[24]. Similar finding was also reported in earlier studies using enzymatic inhibitors of KDM6A/B in liver cancer cells, which proved the importance of these enzymes in maintaining the liver cell identity and proliferation in vitro[27–29]. In contrast, the metabolic function of KDM6A is yet largely unexplored. Because KDM6A expression in tissues differs between males and females, whether such difference may contribute to the sex dimorphism in tissue metabolism remains less known. By integrating transcriptomic and epigenomic techniques in human liver cell lines and liver-specific knockout mice and disease models, we plan to investigate the physiological roles and regulatory mechanisms of KDM6A in the liver. We want to address the following questions: i) what the metabolic function of KDM6A in regulating liver lipid metabolism is; ii) how such regulation is modulated in the chromatin epigenetic levels; iii) whether liver KDM6A deficiency leads to sex difference in developing metabolic disorders.

In this study, we discovered that KDM6A knockdown in cell lines derived from females, but not males induced transcriptional changes annotated to lipoprotein and cholesterol metabolic pathways linked with cardiovascular disorders. In accordance, hepatocyte specific Kdm6a knockout (LKO) mice showed unique atherogenic lipoprotein profiles and were prone to developing atherosclerosis upon genetic and dietary challenges, specifically in females but not in males. Cistrome and epigenome analysis revealed that KDM6A is recruited by Hepatic Nuclear Factor 4 Alpha (HNF4A) to the cis-regulatory regions of lipoprotein and cholesterol metabolic genes such as APOA1 (encoding Apolipoprotein A1). This led to subsequent epigenetic changes which enhanced the binding of cholesterol metabolic TFs such as cAMP-responsive element-binding protein H (CREBH, encoded by CREB3L3) to induce the transcription of lipoprotein and cholesterol metabolic genes in hepatocytes. Interestingly, this KDM6A-HNF4A-CREBH network was also observed to control lipoprotein and cholesterol metabolic genes in human liver cell lines from females but not males.

We have therefore identified a cholesterol regulatory mechanism controlled by X-linked epigenetic enzyme KDM6A in the liver. Deciphering the key hubs for the transcription networks of liver cholesterol and lipoprotein metabolism is not only essential for understanding the personalized drug responsiveness in patients but also for identifying novel targets and strategies for intervention.

## Results
### Hepatocyte-specific depletion of KDM6A triggers lipoprotein dysregulation and atherosclerosis

To determine the function of KDM6A in the liver cells, we first performed siRNA knockdown of KDM6A in the human liver cell line HROHep03 (derived from a female patient, hereafter referred to as HRO). KDM6A knockdown in the HRO cells led to 195 downregulated and 93 upregulated genes (Fig. 1a). To study whether KDM6A plays a similar role in male liver cells, we surveyed transcriptome of various liver cell lines using public available RNAseq dataset[30]. We compared the relative KDM6A and KDM6C mRNA levels and their expression ratio with male human primary hepatocytes, and selected Huh1 cells as a human male liver cell model (Supplementary Fig. 1a). Noteworthy, commonly used cell lines such as HepG2 and Huh7 cells had very low/ no expression of KDM6C mRNA validated by Quantitative PCR (qPCR) (Supplementary Fig. 1b). Short Tandem Repeat (STR) cell line database confirmed that some of the liver cancer cell lines from males such as Huh7 does not contain Y chromosome. We also tested the karyotypes of HRO and Huh1 cells and found that HRO cells had 2 copies of X chromosomes and Huh1 had 1 copy of X chromosome like Huh7 and HepG2 cells (Supplementary Fig. 1c). Knockdown of KDM6A in the Huh1 cells led to many more downregulated (344) than upregulated (55) genes, similar to HRO cells (Fig. 1b). However, Gene ontology (GO) analysis showed that the downregulated genes were annotated to pathways of cholesterol and lipoprotein metabolism in the HRO cells but not Huh1 cells (Fig. 1c). Representative genes in the KDM6A knockdown HRO cells included APOA1 and APOM, both are components of HDL and involved in their protective roles in atherosclerosis[31,32], as well as APOH which was reported to be associated with liver lipid accumulation, steatosis and bile acid regulation[33,34] (Fig. 1d). Such cholesterol and lipoprotein metabolic pathways were not enriched in the upregulated genes in both HRO and Huh1 cells (Supplementary Fig. 1d). Of note, HRO and Huh1 cells had very few common KDM6A regulated genes (Supplementary Fig. 1e). This contrasted with data from the Y-absent Huh7 cells. More than 50% of the downregulated and 20% of the upregulated genes in the HRO cells were overlapped with Huh7 cells upon KDM6A knockdown (Supplementary Fig. 1f). Interestingly, common KDM6A target genes between HRO and Huh7 cells were also annotated to lipoprotein and cholesterol metabolic pathways (Supplementary Fig. 1g). Because the X-linked KDM6A and Y-linked KDM6C share similar structures, the two proteins possibly compensated each other's function in the liver cells. Indeed, several key targets of KDM6A such as APOM and APOH could only be reduced by double knockdown of KDM6A and KDM6C in the

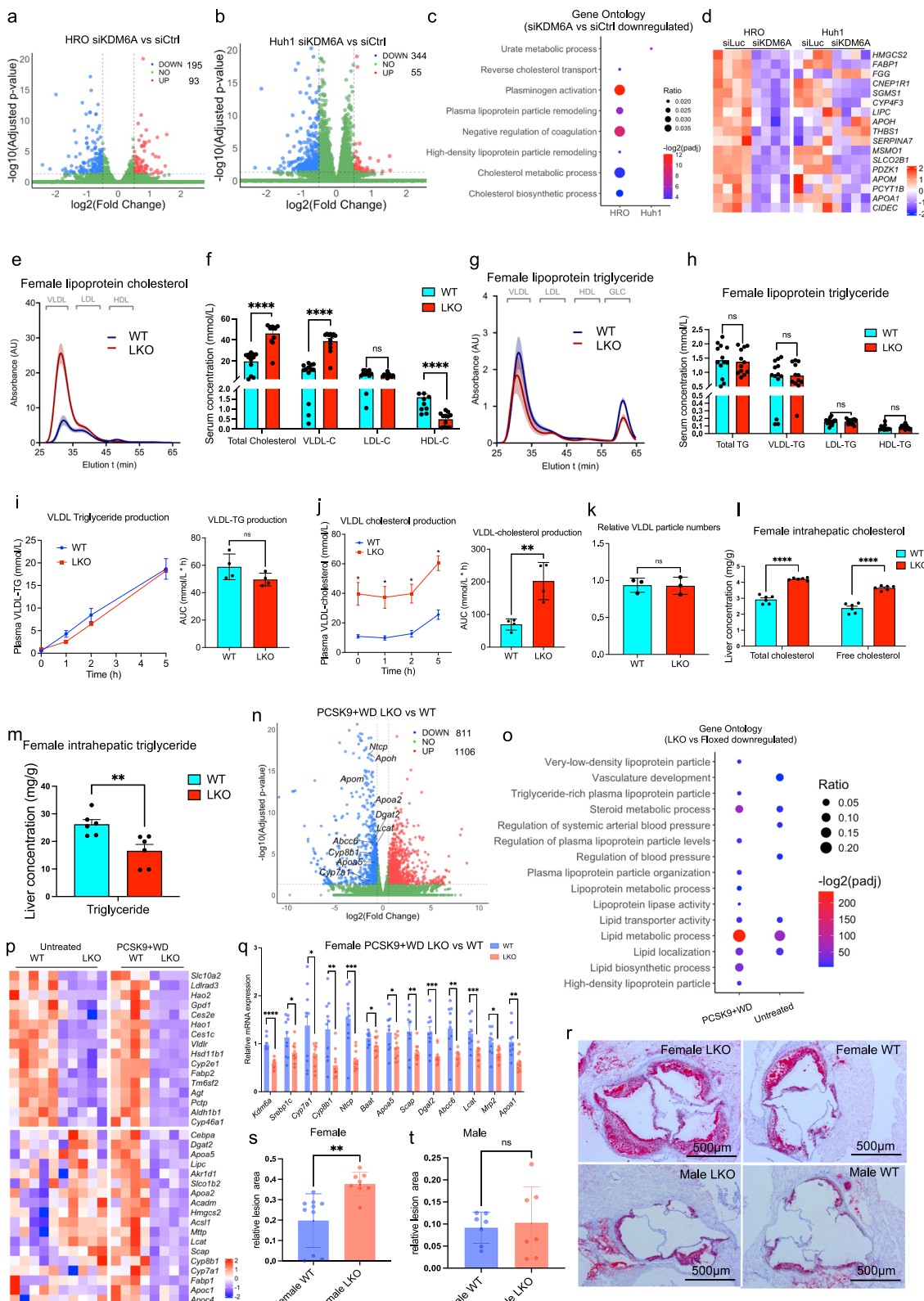

Huh1 cells, supporting the redundant function of KDM6C and KDM6A on those genes (Supplementary Fig. 1h).

We continued to investigate the in vivo function of KDM6A using LKO mice. Removal of KDM6A surprisingly changed very few genes in the male (13 upregulated versus 13 downregulated) comparing to female (560 downregulated versus 429 upregulated) mice (Supplementary Fig. 1i, j). The downregulated genes in the male LKO were also

quite different from those changed in the female LKO mice (Supplementary Fig. 1k). Interestingly, fast protein liquid chromatography (FPLC) analysis of lipoprotein cholesterol and triglyceride revealed elevation of LDL cholesterol but not triglyceride specifically in chow diet fed female but not male LKO mice (Supplementary Fig. 2a–d). Lipoprotein and cholesterol metabolic dysregulation are tightly connected to cardiovascular diseases such as atherosclerosis. To study the

**Fig. 1 | KDM6A ablation leads to transcriptomic and phenotypic changes linked with atherosclerosis. a** Volcano plot of RNA-seq in siKDM6A versus siLuc in HRO cells. *P* values were calculated using the Wald test (two-sided). To account for multiple hypothesis testing, the raw P-values were adjusted using the Benjamini-Hochberg (BH) method. **b** Volcano plot of RNA-seq in siKDM6A versus siLuc in Huh1 cells. *P* values were calculated using the Wald test (two-sided). To account for multiple hypothesis testing, the raw P-values were adjusted using the Benjamini-Hochberg (BH) method. **c** Gene ontology analysis in the decreased genes upon KDM6A knockdown in HRO and Huh1 cells. Statistical significance was determined by a one-tailed hypergeometric test against the background of all protein-coding genes in the human genome. To correct for multiple hypothesis testing, raw P-values were adjusted using the Benjamini-Hochberg (BH) false discovery rate (FDR) method. The size of the dots shows the gene ratio; the color of dots represents adjusted p value of each pathway. **d** Heatmap of downregulated genes upon KDM6A knockdown in HRO and Huh1 cells, representative lipoprotein and cholesterol metabolic genes are listed. **e** FPLC curve and (**f**) quantification of lipoprotein cholesterol levels (Total Cholesterol: $P < 0.000001$, VLDL-C: $P < 0.000001$, LDL-C: $P = 0.635141$, HDL-C: $P = 0.000014$), and (**g**) FPLC curve and (**h**) quantification of triglyceride profiles in AAV-PCSK9 female WT ($n = 13$) and LKO mice ($n = 12$), two-tailed unpaired Student's t test. FPLC analysis of VLDL (**i**) triglyceride (TG) and (**j**) cholesterol ($P = 0.0043$) production at indicated time points upon Poloxamer-407 injection in AAV-PCSK9 mice ($n = 4$ per group), two-tailed unpaired Student's t test. **k** SPP analysis of relative VLDL particle numbers in WT and LKO female mice ($n = 3$ per group), two-tailed unpaired Student's t test. **l** Cholesterol (Total

Cholesterol: $P < 0.000001$, Free cholesterol: $P = 0.000003$) and (**m**) Triglyceride ($P = 0.006972$) levels in the livers of female floxed ($n = 6$) and LKO ($n = 6$) mice, two-tailed unpaired Student's t test. **n** Volcano plot showing the transcriptomic changes female AAV-PCSK9 LKO versus WT mice. *P* values were calculated using the Wald test (two-sided). To account for multiple hypothesis testing, the raw P-values were adjusted using the Benjamini-Hochberg (BH) method. **o** Gene ontology analysis of downregulated genes in female chow diet and AVV-PCSK9 treated mice. The size of the dots represents gene ratio and the color of the dots shows -log2padj. Statistical significance was determined by a one-tailed hypergeometric test against the background of all protein-coding genes in the mouse genome. To correct for multiple hypothesis testing, raw *P* values were adjusted using the Benjamini-Hochberg (BH) false discovery rate (FDR) method. **p** Heatmaps showing transcriptomic changes in untreated and AAV-PCSK9 WT and LKO mice. **q** qPCR analysis of WT ($n = 10$) and LKO ($n = 11$) female AAV-PCSK9 mouse liver biopsies, two-tailed unpaired Student's t test. *Kdm6a*: $P < 0.0001$, *Srebp1c*: $P = 0.0303$, *Cyp7a1*: $P = 0.0210$, *Cyp8b1*: $P = 0.0021$, *Ntcp*: $P = 0.0002$, *Baat*: $P = 0.0397$, *Apoa5*: $P = 0.0383$, *Scap*: $P = 0.0046$, *Dgat2*: $P = 0.0008$, *Abcc6*: $P = 0.0011$, *Lcat*: $P = 0.0008$, *Mrp2*: $P = 0.0121$, *Apoa1*: $P = 0.0011$. **r** Representative oil red staining of aortic roots of female (upper panels) and male (lower panels) mice. Relative lesion area quantification in (**s**) female ($n = 10$ in WT group, $n = 8$ in LKO group)($P = 0.0025$) and (**t**) male($n = 7$ in WT group, $n = 7$ in LKO group) mice is shown, two-tailed unpaired Student's t test. All data are represented as mean ± s.e.m.*$P < 0.05$, **$P < 0.01$, ***$P < 0.001$, ****$P < 0.0001$. Source data are provided as a Source Data file.

function of KDM6A in atherosclerosis, we created an adeno-associated virus (AAV)-driven disease model. The AAV expresses the mouse constitutively active PCSK9 (harboring a variant encoding p.D377Y) driven by the liver-specific *Thyroxine Binding Globulin* (*Tbg*) promoter. The mice were then fed with high fat and high cholesterol diet (western diet-WD) for 12 weeks[35] (hereafter referred to as AAV-PCSK9). The plasma FPLC analysis showed that KDM6A deletion in the liver induced an over 2-fold increase of VLDL cholesterol and a more than 70% reduction of HDL cholesterol without affecting the lipoprotein triglyceride levels (Fig. 1e, f). While the VLDL triglyceride level remained unchanged (Fig. 1g, h). Again, such changes were only observed in female but not in male mice (Supplementary Fig. 2e–h). In order to identify the mechanisms of the increased VLDL cholesterol, we performed a VLDL production assay in the female AAV-PCSK9 WT and LKO mice by inhibiting the lipase activity in the mice with poloxamer-407 (P-407) and measuring the VLDL triglyceride and cholesterol at different time points by FPLC (Fig. 1i, j). We found that VLDL triglyceride production was slightly but not significantly decreased, while VLDL cholesterol level increased by 2 folds (Fig. 1i, j). Analysis of the VLDL fraction revealed a significantly higher cholesterol-to-triglyceride ratio in LKO mice compared to WT controls (Supplementary Fig. 2i), indicating an altered VLDL particle composition in the LKO mice. While total plasma APOB100 levels were elevated in female LKO mice during the VLDL production assay (Supplementary Fig. 2j), these levels remained uncoupled from the progressive rise in VLDL triglycerides and cholesterol following lipase inhibition by P-407 (Fig. 1i, j). Because both APOB100 and APOB48 can be structural components of murine VLDL particles, APOB100 alone cannot reflect the VLDL difference between the WT and LKO mice. To precisely determine VLDL particle numbers, we employed Single-Particle Profiling (SPP)[36]. SPP is a state-of-the-art high-resolution biophysical technique which analyses fluorescently stained purified VLDL using a specific chemical called NR12S[37]. By monitoring the continuous emission signal fluctuation from multiple fluorescent channels, the technique was able to directly quantify the VLDL particle numbers in the WT and LKO mice serum. The results demonstrated that relative VLDL particle numbers were similar between WT and LKO mice (Fig. 1k), supporting that the observed hypercholesterolemia in LKO mice was driven by cholesterol-enrichment of VLDL particles rather than an increase in particle secretion. Interestingly, the intrahepatic total and free cholesterol levels increased by more than 30% while the liver triglyceride

level was significantly lower in the LKO female mice (Fig. 1l, m). In contrast, the male WT and LKO AAV-PCSK9 mice did not show difference (Supplementary Fig. 2k, l). To investigate the transcriptomic regulation by KDM6A in the liver, we performed RNAseq in the WT and LKO AAV-PCSK9 mice. The result showed approximately identical down and upregulated gene number (Fig. 1n). The downregulated genes in both untreated and AAV-PCSK9 mice were annotated to multiple lipoprotein metabolic pathways (Fig. 1o). In particular, we have observed reduced expression of genes in lipid and cholesterol metabolic pathways in the AAV-PCSK9 mice (Fig. 1p, q). For example, reduced bile acid synthesis genes such as *Cyp7a1*, *Cyp8b1* and *Akr1d1* might attenuate liver cholesterol clearance via bile acids and lead to accumulated cholesterol in the liver (Fig. 1q), while downregulation of bile acid process and excretion genes like *Baat*, etc, might further sequest bile acids in the liver which was observed in the AAV-PCSK9 female LKO mice (Supplementary Fig. 2m, n). In addition, total bile acid levels were almost 10 times higher in the LKO females but not male mice (Supplementary Fig. 2o). Bile acid composition analysis in the mouse serum showed that the unconjugated cholic acid (CA), deoxycholic acid (DCA), chenodeoxycholic acid (CDCA) and ursodeoxycholic acid (UDCA) were not different, but the conjugated bile acids such as taurine-CA (TCA), TDCA, TCDCA and TUDCA were more than 10 times higher in the LKO female mice (Supplementary Fig. 2p, q). Moreover, the rodent-specific unconjugated muricholic acids (MCA) were not changed, but the conjugated T-MCAs were also more than 10 times higher in the female LKO mice serum (Supplementary Fig. 2r, s).

To further investigate the impact of KDM6A loss on bile acid homeostasis, we measured plasma levels of 7α-hydroxy-4-cholesten-3-one (C4), a commonly utilized surrogate marker for hepatic bile acid synthesis. Surprisingly, despite the downregulation of the rate-limiting enzyme CYP7A1, female LKO mice exhibited significantly elevated plasma C4 levels (Supplementary Fig. 2t). This elevation likely reflected a complex metabolic uncoupling common in cholestatic liver injury. CYP8B1 is a key enzyme which converts C4 into bile acids. The reduction of this enzyme in the LKO mice might create a metabolic bottleneck, leading to the accumulation and subsequent spillover of C4 into the systemic circulation despite a potential reduction in de novo biosynthesis. Therefore, while the C4 levels are elevated, the marked accumulation of conjugated bile acids and the suppression of transcriptional synthesis markers collectively pointed toward a cholestatic phenotype with impaired bile acid excretion.

These data strongly suggest the imbalanced liver cholesterol metabolism caused by defect bile acid excretion, leading to the conjugated bile acids and intermediates to spill over into the blood. In addition, the transporters SLCO1B2 and NTCP which control bile acid reuptake and were also significantly lower in the LKO mice, impairing the transport of bile acids back to the liver. The reduced liver triglyceride might be caused by lower expression of lipogenesis genes such as *Srebp* and *Dgat2* in the female LKO mice. Some of the genes were tested in the protein level by western blot (Supplementary Fig. 2u). Collectively, *Kdm6a* knockout in the female liver altered multiple cholestearl and lipid metabolic pathways which changed the intrahepatic triglyceride and cholesterol levels. This might in turn affected the liver-derived VLDL triglyceride and cholesterol composition in the AAV-PCSK9 mice. Not surprisingly, the elevated atherogenic lipoprotein profile was paralleled with a more severe atherosclerosis phenotype in the LKO female but not male mice, as shown by the aortic root oil red staining and quantification in the AAV-PCSK9 WT and LKO mice (Fig. 1r–t). Using a public type 2 diabetes (T2D) knowledge portal[38], we explored the association of the rare coding variants in KDM6A with cardiovascular disease traits. We found that KDM6A variants correlated positively with apoA1 and negatively with apoB levels (Supplementary Fig 2v), consistent with what was discovered in the LKO mice (Fig. 1e–h, Supplementary Fig. 2a–d).

## KDM6A regulation of lipoprotein metabolic genes is independent of H3K27me3 demethylation

KDM6A controls histone H3K27me3 demethylation through its demethylase activities[22]. It also works with other coregulator complexes such as MLL3/4 and CBP to modulate histone H3K4me1/2 and H3K27ac, independent of its enzymatic functions[22]. To decipher the epigenetic mechanisms underlying the lipoprotein regulation by KDM6A, we performed chromatin immunoprecipitation sequencing (ChIP-seq) of H3K4me1/2, H3K27ac and H3K27me3 in control and siKDM6A transfected HRO and Huh1 cells. The coverage plots showed that KDM6A knockdown did not change global H3K4me1/2 or H3K27ac, with minor or no effects to increase genome-wide histone H3K27me3 in both HRO and Huh1 cells (Fig. 2a–h). Similarly, in the downregulated genes, HRO and Huh1 cells also had very minor overall changes at H3K4me1/ 2 (Huh1) or H3K27ac (Huh1 and HRO) but not H3K27me3 (Supplementary Fig. 3a–h). However, when we performed correlation analysis by comparing H3K4me1/2, H3K27ac and H3K27me3 epigenetics changes with siKDM6A altered transcriptomics at significantly regulated gene loci (padj<0.05) in both HRO and Huh1 cells, we observed significant (although weak) correlations of H3K4me1/2 and H3K27ac but not H3K27me3 changes with mRNA expression (Fig. 2i–p), Suggesting the transcriptomic regulation by KDM6A did not rely on H3K27me3 in both HRO and Huh1 cells.

We then asked whether KDM6A regulation on its target lipoprotein and cholesterol metabolic genes in HRO cells was also independent of H3K27me3. To study this, we checked all the genes annotated to lipoprotein and cholesterol metabolic pathways, as well as cardiovascular related diseases in the GO and DO analysis, and identified 135 genes in total, hereafter referred to as lipoprotein and cholesterol (LnC) metabolic genes (Supplementary Fig. 3i). We then performed a correlation analysis to compare the H3K4me1/2, H3K27ac and H3K27me3 with the mRNA expression changes specifically at the LnC gene loci. While there was no significant correlation of H3K27me3 with the LnC genes, H3K4me1/2 and H3K27ac were all significantly associated with LnC mRNA expression (Supplementary Fig. 3j–m). This was consistent with epigenetic changes at *APOA1/APOC3* gene cluster, which showed decreased H3K4me1/2 and H3K27ac, but not H3K27me3 upon KDM6A knockdown in the HRO cells (Fig. 2q). Similar findings were observed at *APOM* and *APOH* loci, further supporting that H3K27me3 was not involved in suppressing the LnC gene expression upon KDM6A knockdown in the HRO cells (Supplementary Fig. 3n, o).

It is not surprising that no epigenetic markers were altered at *APOA1/APOC3*, *APOM* or *APOH* were not regulated by KDM6A at Huh1 cells as none of the genes were downregulated upon KDM6A knockdown in the Huh1 cells (Fig. 2r, and Supplementary Fig. 3p, q). However, at Huh1-specific KDM6A target gene loci exemplified by *SLC6A19*, similar reduction of H3K4me1/2 and H3K27ac was observed, while H3K27me3 maintained the same (Fig. 2s). In consistence, knocking down of MLL3/4 (or KMT2C/2D) had similar effects on LnC genes such as *APOA1*, *APOM* and *APOH* in the HRO cells (Fig. 2t).

We also performed H3K4me1/2, H3K27ac and H3K27me3 ChIP-seq in the female floxed and LKO mouse liver samples. In contrast to the HRO cells, there was a slight decrease in H3K4me1 both genome-widely and at downregulated gene loci (Supplementary Fig. 3r, s). The H3K27me3 showed a marginal global decrease in the female LKO liver but not at downregulated gene loci (Supplementary Fig. 3r, s), while H3K4me2 and H3K27ac were not noticeably changed (Supplementary Fig. 3r, s). At *Apoa1/Apoc3/Apoa4* gene cluster, the H3K4me1/2 and H3K27ac were decreased, but H3K27me3 was not changed (Supplementary Fig. 3t) At the *Apom* locus, H3K4me1 and H3K27ac were decreased while H3K27me3 remained consistently unchanged (Supplementary Fig. 3u).

Collectively, our finding suggested that the KDM6A regulation on both global transcriptome and LnC genes in the HRO cells was independent of its H3K27me3 demethylase activities but rather require its non-enzymatic function via interplaying with complexes such as MLL3/4.

## HNF4A is required to recruit KDM6A to lipoprotein genes and regulates their transcription

KDM6A binding at HRO and Huh1 cells appeared largely overlapped (Supplementary Fig. 4a). To identify the TFs responsible for KDM6A recruitment to the LnC genes, we first did a motif analysis in KDM6A CUT&Tag peaks in HRO cells. Such analysis has identified multiple TF motifs at KDM6A binding sites genome widely (Fig. 3a). KDM6A CUT&Tag peaks in Huh1 cells were also enriched with identical top motifs as in HRO cells (Supplementary Fig. 4b), with NF4A-DR1 as common motifs with HepG2 KDM6A ChIP-seq peaks (Supplementary Fig. 4c). To further narrow down the TF candidates, we performed a rapid immunoprecipitation mass spectrometry (RIME)[39], in which we immunoprecipitated KDM6A in the chromatin of Huh7 nuclei for proteomics analysis. The RIME experiment unbiasedly identified multiple TFs interacting with KDM6A in the chromatin level (Fig. 3b). Integrating the motif analysis with the RIME results identified HNF4A as a key transcription factor. We then compared existing HNF4A ChIP-seq[40] with the KDM6A ChIP-seq[40] results in the HepG2 cells, and found that more than 90% of KDM6A peaks were colocalized with HNF4A (Supplementary Fig. 4d).

To further demonstrate that HNF4A is required for KDM6A binding in the human liver cells, we performed CUT&Tag analysis of KDM6A in control versus HNF4A knockdown HRO and Huh1 cells. KDM6A was also genome-widely decreased upon HNF4A knockdown in HRO cells (Fig. 3c, d). There were 8546 KDM6A peaks decreased upon HNF4A knockdown in the HRO cells (Fig. 3e). These peaks are annotated to LnC genes such as *APOA1/4*, *APOM* and *APOH*, among many others (Fig. 3e). To be noted, at *APOA1/APOC3* and *APOA5* and *APOC2/4* loci which had both KDM6A and HNF4A peaks. When HNF4A was removed, the KDM6A binding almost disappeared at these gene locus (Fig. 3f, and Supplementary Fig. 4e–f). In contrast, *CLPTM1* gene promoter only had KDM6A peak without HNF4A, and knockdown of HNF4A did not affect KDM6A at this locus (Supplementary Fig. 4f).

In contrast, knockdown of HNF4A in Huh1 cells did not change global KDM6A binding (Supplementary Fig. 4g–i). However, at some LnC gene loci such as *APOA1/APOC3* cluster, siHNF4A released KDM6A binding in the Huh1 cells (Supplementary Fig. 4j). This seemingly contradicted result might be explained by coregulation of such LnC

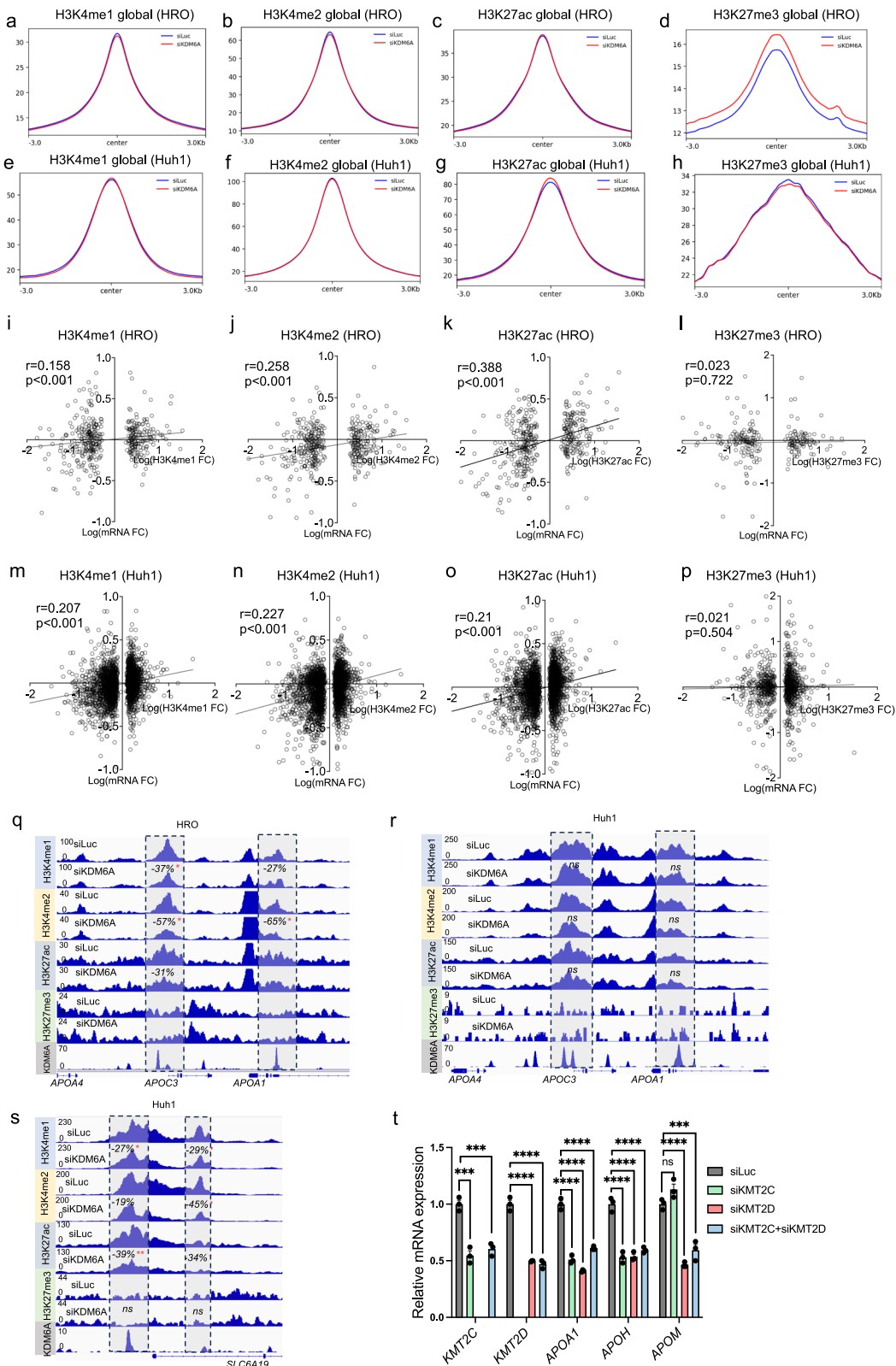

genes by both X-lined KDM6A and Y-linked KDM6C. As both enzymes share similar transcription factor binding domains, they are likely to be recruited by HNF4A at *APOA1/APOC3* loci. Loss of HNF4A therefore resembled KDM6A and KDM6C double knockdown at those genes, leading to reduced expression of *APOA1/APOC3* and released KDM6A at the same time (Supplementary Fig. 1h), while KDM6A knockdown alone did not change *APOA1/APOC3* mRNA in the Huh1 cells (Fig. 1d).

The coregulation of LnC genes by KDM6A and HNF4A was further evaluated by comparing the transcriptomic changes in KDM6A and HNF4A knockdown HRO cells separately. Despite that there were only 80 commonly regulated genes in comparison to 1368 and 206 HNF4A and KDM6A-specific genes, there was a significant correlation between mRNA changes siKDM6A and siHNF4A HRO cells (Fig. 3g, and Supplementary Fig. 4k). LnC genes such as *APOA1 and APOM* were targets of both HNF4A and KDM6A (Fig. 3g). GO analysis in the KDM6A/HNF4A

**Fig. 2 | KDM6A regulated lipoprotein and cholesterol metabolic pathways are independent of its enzymatic activities.** Global (**a**) H3K4me1, **b** H3K4me2, **c** H3K27ac and (**d**) H3K27me3 changes upon KDM6A knockdown in HRO cells. Global (**e**) H3K4me1, **f** H3K4me2, **g** H3K27ac and (**h**) H3K27me3 changes upon KDM6A knockdown in Huh1 cells. Correlation of KDM6A-regulated gene expression with (**i**) H3K4me1 ($P < 0.001$), **j** H3K4me2 ($P < 0.001$), (**k**) H3K27ac ($P < 0.001$) and (**l**) H3K27me3 changes in HRO cells. Correlation of KDM6A-regulated gene expression with (**m**) H3K4me1 ($P < 0.001$), **n** H3K4me2 ($P < 0.001$), **o** H3K27ac ($P < 0.001$) and (**p**) H3K27me3 changes in Huh1 cells. Correlation between KDM6A-regulated gene expression changes (log2 fold change) and changes in each histone modification (log2 fold change) was assessed using Pearson correlation coefficient. For each modification (H3K4me1, H3K4me2, H3K27ac, and H3K27me3), the Pearson's r and its statistical significance were calculated using a two-tailed t-test. $P < 0.05$ was considered statistically significant. Genome browser screenshot of H3K4me1, H3K4me2, H3K27ac and H3K27me3 ChIP-seq in siLuc and siKDM6A (**q**) HRO and (**r**) Huh1 cells at *APOA1/APOC3/APOA4* gene cluster, and (**s**) *SLC6A19* gene locus in Huh1 cells. (merged bigwig files from $n = 4$ in each group). **t** qPCR analysis of key lipoprotein genes in siKMT2C, siKMT2D and combined knockdown HRO cells. ($n = 3$ biological replicates in each group), one-way ANOVA followed by Tukey's test. (KDM2C: siLuc vs. siKMT2C Adjusted *P* Value = 0.0003, siLuc vs. siKMT2C+siKMT2D Adjusted *P* Value = 0.0006. KDM2D: siLuc vs. siKMT2D Adjusted *P* Value < 0.0001, siLuc vs. siKMT2C+siKMT2D Adjusted *P* Value < 0.0001. APOH: siLuc vs. siKMT2C Adjusted *P* Value < 0.0001, siLuc vs. siKMT2D Adjusted *P* Value < 0.0001, siLuc vs. siKMT2C+siKMT2D Adjusted *P* Value < 0.0001. APOA1: siLuc vs. siKMT2C Adjusted *P* Value < 0.0001, siLuc vs. siKMT2D Adjusted *P* Value < 0.0001, siLuc vs. siKMT2C+siKMT2D Adjusted *P* Value < 0.0001. APOM: siLuc vs. siKMT2D Adjusted *P* Value < 0.0001, siLuc vs. siKMT2C+siKMT2D Adjusted *P* Value = 0.0003.) All data are represented as mean ± s.e.m.**P* < 0.05, ***P* < 0.01, ****P* < 0.001, *****P* < 0.0001. Source data are provided as a Source Data file.

coregulated as well as KDM6A or HNF4A specific genes demonstrated that the commonly downregulated genes upon KDM6A and HNF4A knockdown were annotated to multiple lipoprotein regulatory pathways (Fig. 3h). The dependency of KDM6A on HNF4A in regulating key LnC genes such as *APOA1*, *APOM* and *APOH* was further proved by double knocking down of KDM6A and HNF4A in the HRO cells, which showed no further decrease of these genes comparing with the siHNF4A group alone (Fig. 3i–j).

We also did similar comparison in Huh1 cells. Despite that KDM6A had very different gene targets in Huh1 cells, with very few lipoprotein and cholesterol metabolic genes, knockdown of KDM6A had a significant correlation with siHNF4A in mRNA expression changes (Supplementary Fig. 4l). The KDM6A and HNF4A commonly regulated genes were not annotated to any cholesterol or lipoprotein metabolic pathways (Supplementary Fig. 4m and 4n). Interestingly, KDM6A was significantly released by HNF4A knockdown in KDM6A/HNF4A common genes in the HRO cells (Supplementary Fig. 4o) but not in Huh1 cells (Supplementary Fig. 4p).

These above data collectively showed that HNF4A and KDM6A interact in the chromatin level to control gene expression in both HRO and Huh1 cells. However, the common target genes were sex specific and mostly annotated to lipoprotein and cholesterol metabolic pathways in the HRO liver cells from female patients.

## The interplay of HNF4A and KDM6A regulates epigenetic changes in the liver cells

We asked whether the chromatin interplay of HNF4A and KDM6A regulated epigenetic changes in the liver cells. Knockdown of HNF4A in the HRO cells had minor changes of H3K4me1 and H3K27me3, and identical H3K4me2 and H3K27ac levels (Fig. 4a–d). While in Huh1 cells, siHNF4A drastically reduced H3K27me3 without changing H3K4me1/2 or H3K27ac (Supplementary Fig. 5a–d), suggesting HNF4A might interact with other epigenetic enzymes such as histone methyltransferases. However, H3K4me1 (Fig. 4e), H3K4me2 (Fig. 4f), H3K27ac (Fig. 4g) and H3K27me3 (Fig. 4h) epigenetic alterations in HNF4A and KDM6A knockdown HRO cells showed significant correlations, similar to observations in Huh1 cells (Supplementary Fig. 5e–h), supporting the concept of coregulation of KDM6A and HNF4A in the control of global epigenetic landscapes. We also analyzed the same epigenetic markers specifically at the HNF4A dependent KDM6A binding sites, the KDM6A peaks which were downregulated upon HNF4A knockdown (Fig. 3e). Only H3K4me1 (Fig. 4i), but not H3K4me2, H3K27ac or H3K27me3 (Fig. 4j–l), was decreased upon HNF4A siRNA transfection in the HRO cells. At *APOA1*, *APOH* and *APOC2/APOC4* gene clusters, H3K4me1 and H3K27ac or H3K4me2 were consistently downregulated, while H3K27me3 remained unchanged in these gene loci (Fig. 4m–o). Interestingly, in the Huh1 cells, knockdown of HNF4A at HNF4A-dependent KDM6A peaks led to reduction of all the epigenetic markers including

H3K4me1/2, H3K27ac and H3K27me3 (Supplementary Fig. 5i–l), indirectly supporting that KDM6A loss might be related with the H3K4me1/2 and H3K27ac changes. We have also seen reduced H3Kme1/2 or H3K27ac levels at *APOA1*, *APOH* and *APOC2/APOC4* clusters (Supplementary Fig. 5m-5o), consistent with KDM6A (and maybe also KDM6C) loss at those loci (Supplementary Fig. 4j).

## KDM6A removal affects the binding and transcription activation of CREBH

We hypothesized that KDM6A/HNF4A-modulated changes of the histone modifications H3K4me1/2 and H3K27ac could affect the chromatin accessibility of TFs at key LnC genes. To identify these TFs, we first performed the TF enrichment analysis using CHEA3[41], which takes into account TF–gene co-expression from RNA-seq studies, TF–target associations from ChIP-seq experiments, and TF–gene co-occurrence computed from crowd-submitted gene lists. Using the coregulated gene list by KDM6A and HNF4A in the RNA-seq data from HRO and Huh1 cells, we have identified very different common TFs between HRO and Huh1 cells (Fig. 5a, and Supplementary Fig. 6a). The common top candidates in HRO cells include CREBH and NR1H4 (FXR), both are key regulators of lipid and cholesterol in the liver. While in Huh1 cells, the enriched top listed TFs were a group of Zinc Finger Proteins (ZNF) family TFs such as ZNF692, etc (Supplementary Fig. 6a). This was not unexpected considering the major transcriptomic difference between HRO and Huh1 cells (Supplementary Fig. 1e) and that there were very few lipid or cholesterol metabolic genes in HNF4A/KDM6A commonly regulated genes (Supplementary Fig. 4m, n). We then tested the response of both mice and HRO cells to FXR agonists GSK2324 and GW4064 upon KDM6A removal. However, most of the genes annotated in the NR1H4 targets (including the lipoprotein modulating *APOA1*, *APOM* or *APOH*) were not induced by FXR agonists, neither in mice (Supplementary Fig. 6b) nor in HRO cells (Supplementary Fig. 6c). Among the very few genes which were stimulated by FXR agonists, *Small Heterodimer Partner* (*SHP*) expression was decreased by KDM6A deletion in both mice and HRO cells (Supplementary Fig. 6b, c). However, SHP is a corepressor which inhibits LXR activities and bile acid synthesis, which leads to cholesterol accumulation in the liver[42]. The reduced SHP in the liver therefore was less likely to contribute to the higher intrahepatic cholesterol and blood LDL cholesterol levels in the LKO mice. In mice, the bile acid export pump *Bsep* expression was reduced by KDM6A knockout, but its alteration was marginal (Supplementary Fig. 6b). We also tested LXR target genes in the mice and HRO cells. Although LXR was not identified in the TF enrichment analysis, analysis by qPCR in the AAV-PCSK9 injected mice showed reduced mRNAs for *Cyp7a1* and *Cyp8b1*, both LXR-regulated rate-limiting enzymes in bile acid synthesis (Fig. 1q). However, treatment with the LXR agonists GW3965 and T0901317 did not induce expression of either gene in the mouse hepatocytes or HRO cells (Supplementary Fig. 6d–e), despite that knockout of KDM6A showed

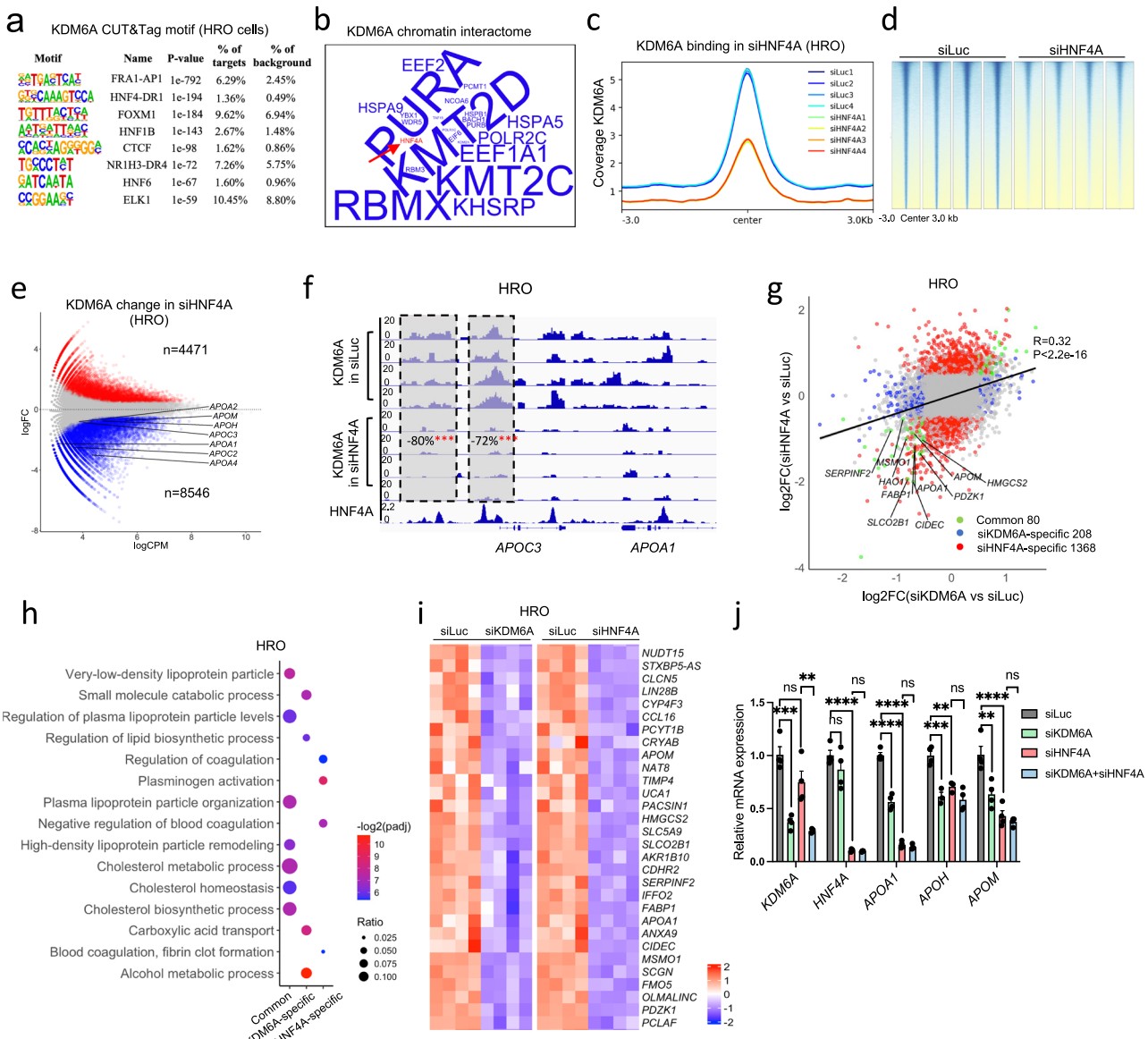

**Fig. 3 | KDM6A interplays with HNF4A to control lipoprotein and cholesterol metabolic genes. a** Motif analysis of KDM6A CUT&Tag peaks in the HRO cells, *P* Values were calculated using cumulative binomial distribution (findMotifsGen-ome.pl in HOMER). **b** Word cloud illustration of interacting proteins with KDM6A in Huh7 cells identified by RIME (*n* = 2 for IgG control and *n* = 2 for KDM6A RIME). The size of the words represents the number of unique peptides. **c** Coverage plot and (**d**) heatmap of the genome wide KDM6A CUT&Tag peak changes upon HNF4A knockdown in HRO cells (*n* = 4). **e** MA plot of the KDM6A CUT&Tag peaks in siHNF4A versus siLuc transfected HRO cells (*n* = 4). The peaks annotated to representative LnC genes are labelled in the plot. **f** Genome browser screenshot of KDM6A in siLuc and siHNF4A knockdown HRO cells aligned with HNF4A ChIP-seq peaks (GSM935619) at *APOA4/APOC3/APOA1* gene cluster. **g** Correlation analysis of RNAseq between KDM6A (x axis) and HNF4A (y axis) knockdown in HRO cells (*P* < 0.0001). The blue, red and green dots represent KDM6A- and HNF4A-specific and commonly regulated genes, non-parametric two-sided Spearman's test. **h** Gene ontology analysis of KDM6A- and HNF4A-specific and commonly downregulated

genes in HRO cells. Statistical significance was determined by a one-tailed hyper-geometric test against the background of all protein-coding genes in the mouse genome. To correct for multiple hypothesis testing, raw *P* values were adjusted using the Benjamini-Hochberg (BH) false discovery rate (FDR) method. The size of the dots means gene ratio and the color shows -log2(padj). **i** Heatmap showing the representative KDM6A and HNF4A commonly regulated genes in HRO cells. **j** qPCR analysis of key LnC genes in siKDM6A, siHNF4A and dual knockdown HRO cells (*n* = 4 biological replicates in each group), one-way ANOVA followed by Tukey's test. (KDM6A: siLuc vs. siKDM6A Adjusted *P* Value = 0.0001, siHNF4A vs. siKDM6A +siHNF4A Adjusted *P* Value = 0.0018. HNF4A: siLuc vs. siHNF4A Adjusted *P* Value < 0.0001. APOA1: siLuc vs. siKDM6A Adjusted *P* Value < 0.0001, siLuc vs. siHNF4A Adjusted *P* Value < 0.0001. APOH: siLuc vs. siKDM6A Adjusted *P* Value = 0.0005, siLuc vs. siHNF4A Adjusted *P* Value = 0.0036. APOM: siLuc vs. siKDM6A Adjusted *P* Value = 0.0028, siLuc vs. siHNF4A Adjusted P Value < 0.0001. All data are represented as mean ± s.e.m.*P* < 0.05, **P* < 0.01, ***P* < 0.001, ****P* < 0.0001. Source data are provided as a Source Data file.

reduced expression of *Cyp7a1* and *Cyp8b1* already in the untreated mouse hepatocytes (Supplementary Fig. 6d). Based on these results, we conclude that the FXR or LXR are not the main regulators directly affected by KDM6A.

We therefore continued to study CREBH from the top TF candidate list. We found that knockdown of CREBH in the HRO cells decreased the expression of key lipoprotein genes such as *APOA1*,

*APOC2*, *APOH* and *APOM* (Fig. 5b), similar to the changes observed in KDM6A knockdown. We also performed ChIP-seq in the control and KDM6A knockdown HRO cells. At the *APOA1* and *APOC2/APOC4* gene clusters, CREBH binding was reduced by 62% and 56% after KDM6A removal (Fig. 5c, d). The changes at *APOH* and *APOM* were not con-clusive because the CREBH peaks at these two gene loci were not clear due to relatively low quality CREBH antibodies. While in KDM6A absent

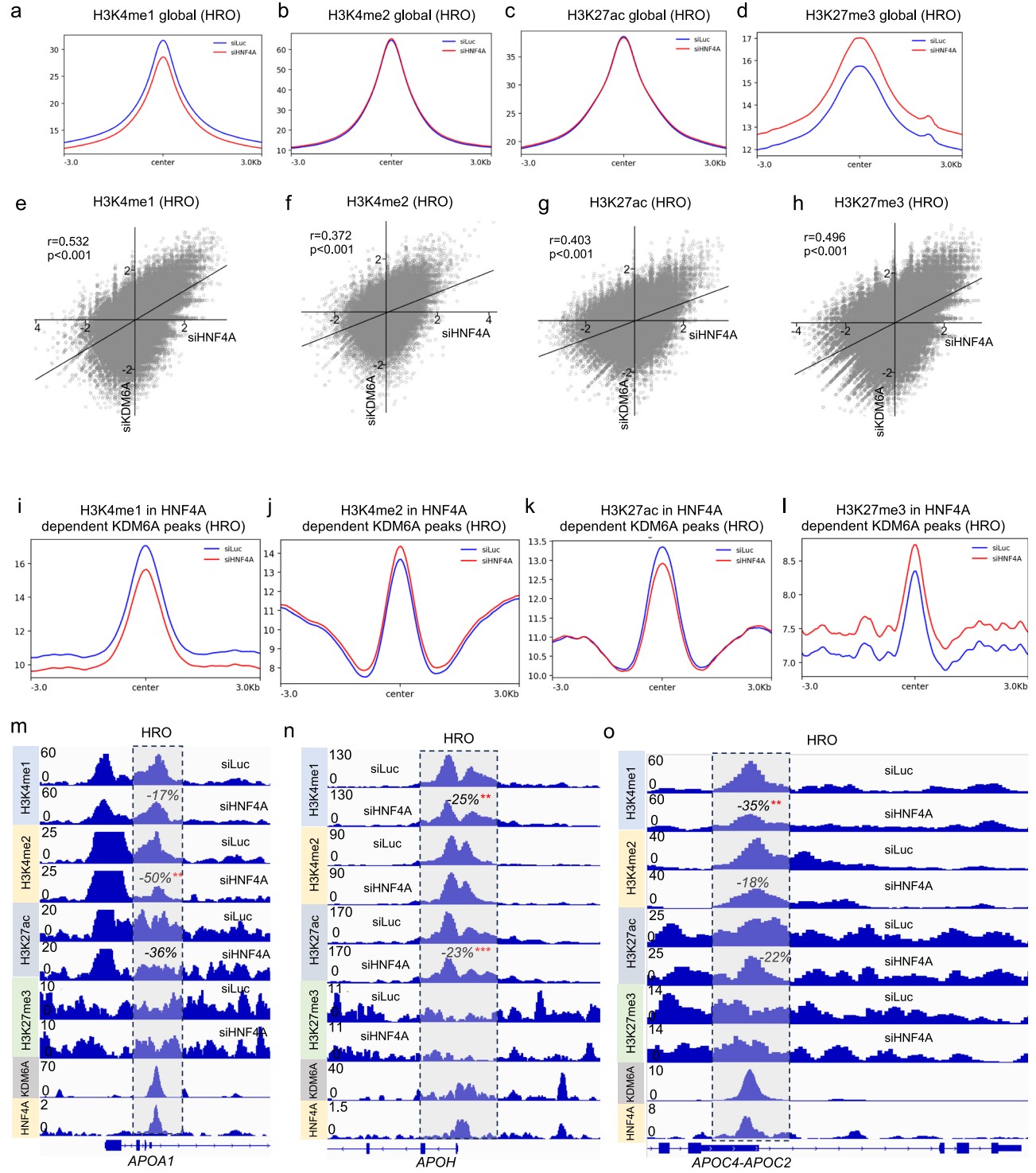

**Fig. 4 | HNF4A coregulated epigenetic changes with KDM6A in the liver cells.**
Coverage plots of (**a**) H3K4me1, (**b**) H3K4me2, (**c**) H3K27ac and (**d**) H3K27me3 in siLuc and siHNF4A transfected HRO cells. Correlation analysis (**e**) H3K4me1 ($P < 0.0001$), (**f**) H3K4me2 ($P < 0.0001$), (**g**) H3K27ac ($P < 0.0001$) and (**h**) H3K27me3 ($P < 0.0001$) ChIP-seq changes in siKDM6A and siHNF4A transfected HRO cells x and y axis represents log2(fold change) of each peak. Correlation was assessed using Pearson correlation coefficient. For each modification (H3K4me1,

H3K4me2, H3K27ac, and H3K27me3), the Pearson's r and its statistical significance were calculated using a two-tailed t-test. $P < 0.05$ was considered statistically significant. Coverage plots of (**i**) H3K4me1, (**j**) H3K4me2, (**k**) H3K27ac and (**l**) H3K27me3 in HNF4A dependent KDM6A peaks. Genome browser screenshots of H3K4me1, H3K4me2, H3K27ac and H3K27me3 ChIP-seq at (**m**) *APOA1*, (**n**) *APOH* and (**o**) *APOC2/APOC4* gene cluster in siLuc and siHNF4A transfected HRO cells (merged bigwig files from $n = 4$ in each group).

gene loci, such as *XR_001752718.3* and *CHRNE*, CREBH binding was not affected in the knockdown group, supporting the validity of the results (Supplementary Fig. 6f, g). The genome-wide recruitment of CREBH was not changed upon KDM6A knockdown in the HRO cells (Fig. 5e), but CREBH binding at KDM6A target loci (KDM6A regulated genes

with KDM6A binding) was significantly reduced upon KDM6A knockdown (Fig. 5f).

We also performed similar analysis in the female AAV-PCSK9 mice. The TF enrichment analysis also identified CREBH and NR1H4 in the top list (Fig. 5g). We also managed to perform CREBH ChIP-seq in

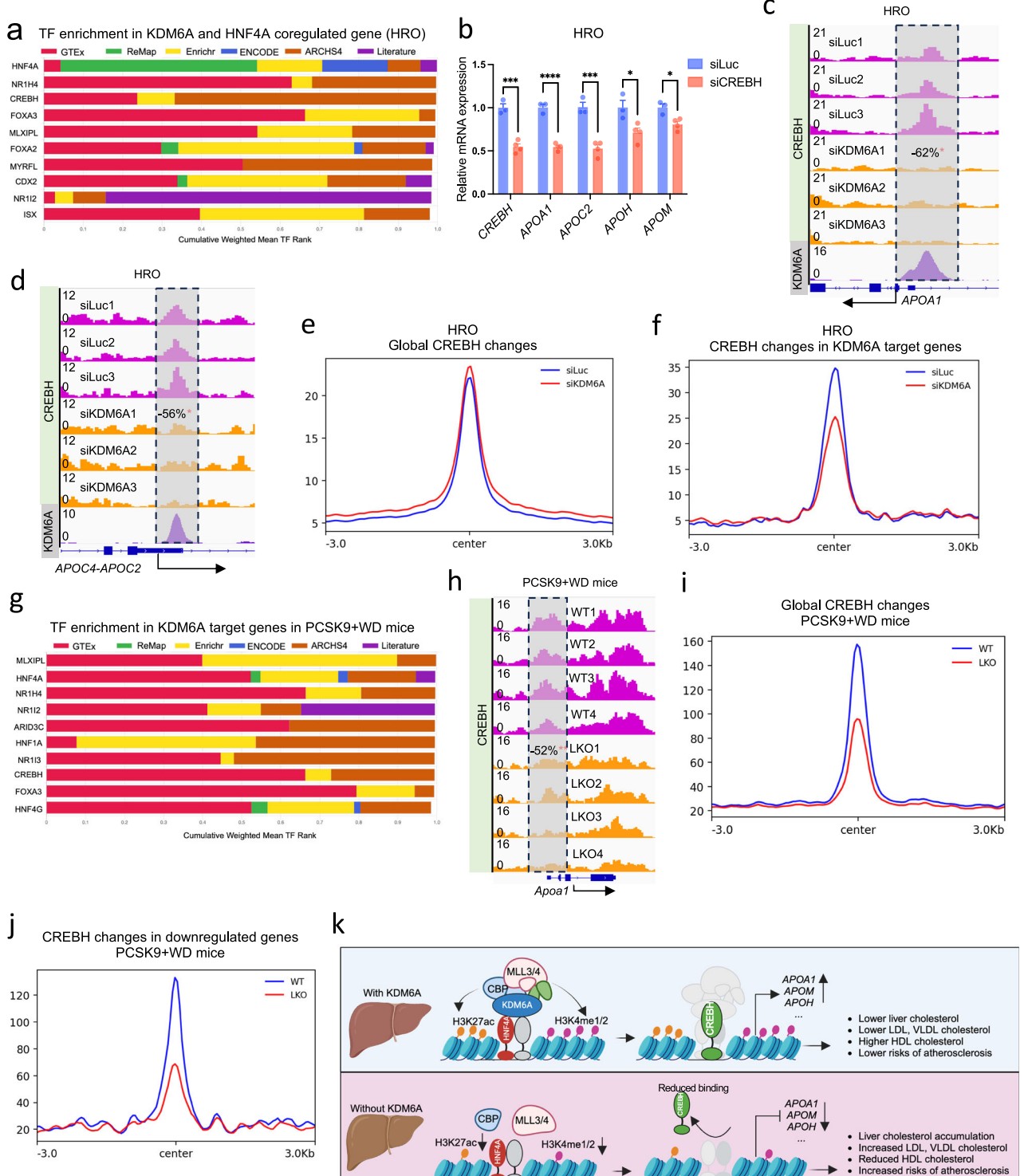

**Fig. 5 | KDM6A and HNF4A coregulated lipoprotein and cholesterol metabolic genes are associated with CREBH. a** ChEA3 analysis of KDM6A and HNF4A coregulated genes in HRO cells. **b** qPCR analysis of key LnC genes upon CREBH knockdown in HRO cells (*n* = 3 in siLuc and *n* = 4 in siCREBH), two-tailed unpaired Student's t test. (CREBH: *P* = 0.0005, APOA1: *P* < 0.0001, APOM: *P* = 0.0146, APOH: *P* = 0.0257, APOC2: *P* = 0.001) Genome browser screenshot of CREBH ChIP-seq peaks in siLuc and siKDM6A transfected HRO cells at (**c**) *APOA1* and (**d**) *APOC2/APOC4* gene cluster (*n* = 3 in each group). Coverage plots showing CREBH ChIP-seq peak intensity in siLuc and siKDM6A transfected HRO cells (**e**) genome widely and (**f**) at KDM6A colocalized peaks at KDM6A target genes (*n* = 3 in each group). **g** ChEA3 analysis of Kdm6a regulated genes in AAV-PCSK9 female LKO

mice. CREBH ChIP-seq peaks in female WT and LKO AAV-PCSK9 mice at (**h**) *Apoa1* gene cluster (*n* = 4 in each group). Coverage plots showing CREBH ChIP-seq peak intensity in female WT and LKO PCSK9 mice (**i**) globally and (**j**) at peaks of down-regulated genes (*n* = 4 in each group). **k** The model illustrating that KDM6A interplays with HNF4A to control epigenetic changes at the lipoprotein and cholesterol metabolic gene loci. This subsequently alters the chromatin microenvironment and affects CREBH binding and transcription of those genes. As a result, the atherogenic blood lipoprotein cholesterol level is increased which is associated with increased risk of cardiovascular diseases. Created in BioRender. Chen, L. (https://BioRender.com/0fe0tkx). All data are represented as mean ± s.e.m.*P < 0.05, **P < 0.01, ***P < 0.001, ****P < 0.0001. Source data are provided as a Source Data file.

mouse liver for the first time. The results showed reduction of CREBH peaks at KDM6A target genes in the mice, such as *Apoa1* (Fig. 5h), *Apoa5*, *Cyp7a1*, *Cyp8b1*, *Lcat*, *Srebf1* and *Dgat2* (Supplementary Fig. 6h–m). While in control gene *Nlrp9b* which was not regulated by *Kdm6a* knockout, CREBH binding was not affected (Supplementary Fig. 6n). We have also found that the CREBH binding in the mouse liver was decreased in the LKO mice both genome-widely (Fig. 5i) and at downregulated gene loci (Fig. 5j) in the AAV-PCSK9 LKO mice.

Based on these results, we propose the model that KDM6A interplays with HNF4A to control the H3K4me1/2 and H3K27ac at lipoprotein and cholesterol metabolic gene loci in the liver cells. These epigenetic changes were linked with chromatin accessibility of CREBH to such loci. As a result, liver deletion of KDM6A affected the binding of CREBH which subsequently reduced the expression of lipoprotein and cholesterol metabolic genes such as *APOA1*, *APOM* and *APOH*. These in combination contributed to the altered lipoprotein cholesterol levels and promoted atherosclerosis in the LKO mice (Fig. 5k).

## Discussion

Our study has identified a previously uncharacterized mechanism by which the epigenetic modulator KDM6A regulates hepatic lipoprotein and cholesterol metabolism, impacting atherosclerosis development. *Kdm6a* LKO female mice showed elevated LDL cholesterol levels in the chow diet conditions and had significantly elevated VLDL and decreased HDL cholesterol upon AAV-PCSK9 and western diet challenges. Mechanistically, KDM6A was recruited by HNF4A to the lipoprotein and cholesterol metabolic genes, creating an active epigenetic microenvironment at the enhancers and promoters of its target genes. This subsequently affected the binding of CREBH and transcription at these gene loci. As a result, *Kdm6a* LKO female mice showed more severe phenotypes of atherosclerosis upon AAV-PCSK9 and western diet treatment. The results were consistent with a previously published association analysis in the females, indicating that DNA methylation levels at the KDM6A promoter is associated with gender-specific HDL levels in humans[43]. We also found that rare variants of KDM6A were associated with higher blood apoB and lower apoA1 levels in humans, supporting the species-conserved role of KDM6A in regulating cholesterol and lipoprotein metabolism in the liver.

*APOA1*, *APOM* and *APOH* were among the KDM6A target genes conserved in both mice and human cells. Both APOA1 and APOM are components of HDL[31,32]. APOA1 interacts with ATP-binding cassette transporters ABCA1 and ABCG1, and is essential for the cholesterol reverse transport from peripheral tissues into the liver by executed by HDL[31]. While APOM was reported to regulate the production of nascent pre-β-HDL, its presence also enhances the cholesterol mobilizing efficacy of HDL[44–46]. The reduced expression of *Apoa1* and *Apom* likely contributes to the decreased HDL cholesterol levels observed in LKO mice, impairing cholesterol efflux from peripheral tissues. While the function of APOH is less known, earlier studies have shown a close link of APOH with liver triglyceride and bile acid metabolism[33,34].

The changes in VLDL were more drastic in the AAV-PCSK9 female LKO mice. VLDL cholesterol was increased by more than 2 folds while VLDL triglyceride levels were slightly decreased in the LKO female mice. These changes were not due to altered VLDL production but was a result of altered VLDL composition. The VLDL cholesterol to triglyceride ratio is much higher in the LKO female mice while the VLDL particle numbers remained the same. Transcriptomic analysis between the AAV-PCSK9 LKO versus WT female mice revealed downregulation of multiple lipid and cholesterol metabolic genes linked with accumulated cholesterol but reduced triglyceride in the female LKO livers, which likely changed the composition of liver-derived VLDL subsequently. Of note, the conjugated bile acid levels in the female LKO mouse serum were more than 10 times higher than the WT. The bile acid levels in the liver were also significantly higher in the female LKO mice, suggesting defect in bile acid excretion. This was aligned with

reduced expression of multiple bile acid efflux genes. Sequestrated bile acid together with the cholesterol promoted liver injury characterized by significant upregulation of a group of inflammatory and fibrosis markers in the female LKO liver, suggesting cholestasis condition. It is unclear whether bile acid synthesis contributed to such pathology. CYP7A1 is the key enzyme involved in bile acid synthesis. Its expression was decreased in the female LKO liver. The reduced *Cyp7a1* could be due to decreased CREBH binding in its locus, but it could also be that the high bile acid level in the female LKO mice repressed *Cyp7a1* expression through secondary feedback loops (i.e., via FXR activation by bile acids). Therefore, it is unclear whether the reduced *Cyp7a1* was the cause or consequence of the elevated bile acids in the liver. On the other hand, C4 is a surrogate marker for liver bile acid synthesis, but its levels were significantly higher in LKO mice. These seemingly contradicted results might be explained by reduced Cyp8b1, which is a key enzyme converting C4 to bile acids, leading to the accumulation of C4 in the liver. The liver injury further induced the spillover of bile acids and intermediates into the blood, raising C4 levels.

The female-specific atherosclerosis lipoprotein profiles in the LKO mice were aligned with the in vitro data. siKDM6A downregulated genes in the HRO cells were annotated largely to lipoprotein and cholesterol metabolism, in contrast to Huh1 cells which showed very different pathways. This might be caused by potential compensatory mechanisms in males. The Y-linked homolog, KDM6C/UTY, is a plausible candidate for mediating this compensation, warranting further investigation into its role in hepatic cholesterol regulation. Both proteins share more than 80% similarity, while KDM6C has significantly lower demethylase activity. In comparison, the coregulator interaction tetratricopeptide repeat (TPR) domains of KDM6A and KDM6C are highly conserved[22]. Therefore, the regulatory mechanisms of KDM6C might to some extent, reflect part of the enzymatic independent functions of KDM6A. Whether KDM6C controls similar cholesterol and lipoprotein modulating pathways as KDM6A in the liver is so far unclear. However, our data showed that knocking down of both KDM6A and KDM6C, but not each of them alone reduced the expression of key lipoprotein genes such as *APOM* and *APOH* (Supplementary Fig. 1h), supporting the potential redundant function of both enzymes in liver cholesterol metabolism. Of note, some studies have reported identical tumor suppressive functions of KDM6A and 6 C in several cancers[26,47]. How the two sex-chromosome linked epigenetic enzymes control transcription in different disease contexts remains an important question to further investigate. In our study, knockdown of KDM6A induced very marginal H3K27me3 changes in the liver cells, the effects on its key target genes such as *APOA1*, *APOM* and *APOH* also seemed to be independent of H3K27me3 but more influenced by H3K4me1/2 and H3K27ac. These results suggest that the cholesterol and lipoprotein modulating functions of KDM6A might require more of its role as an adaptor, via interaction with other coregulator complexes, such as Complex of Proteins Associated with Set1 (COMPASS) or CBP in the HRO cells from females. Interestingly, despite the major difference of KDM6A target genes in HRO and Huh1 cells, mRNA changes in siKDM6A Huh1 cells also showed significant correlation with H3K4me1/2 and H3K27ac, but not H3K27me3, further indicating that the non-enzymatic function of KDM6A also played an important role in regulating its target gene expression in liver cells from male patients. How KDM6A and KDM6C contribute to the sex dimorphism in different disease contexts remains to be further explored.

We have also identified HNF4A as a potential interactive partner of KDM6A in regulating the chromatin level through unbiased cistromic and proteomic methods. HNF4A is a crucial liver lineage determination factor essential for liver development. Liver specific *Hnf4a* mice showed some similar phenotypes as *Kdm6a* LKO mice. Both strains had increased liver cholesterol accumulation (partially due to decreased

*Cyp7a1* expression and bile acid synthesis)[48], and reduced HDL cholesterol (due to decreased expression of *Apoa1/2*)[49]. The VLDL profile changes between *Hnf4a* LKO and *Kdm6a* LKO mice were different because HNF4A regulates a group of VLDL assembling genes such as *Apob*, and knockout of *Hnf4a* led to a much stronger reduction of *Mttp* (similar to *Kdm6a* LKO which also shows a mild decrease)[50], therefore *Hnf4a* LKO sequestrated the VLDL in the liver, leading to accumulation of both intrahepatic triglyceride and cholesterol but reduced blood VLDL triglyceride and cholesterol levels[49]. Apart from that, Hnf4a but not Kdm6a controls expression of *Ldlr*[49], which is a major regulator of blood LDL cholesterol. Overall, *Hnf4a* LKO has a much stronger phenotype comparing to *Kdm6a* LKO. From our RNA-seq comparison results in HRO cells (Supplementary Fig. 4k), we have also observed a much broader regulatory targets of HNF4A comparing to KDM6A. This is also supported by the ChIP-seq results that the binding of HNF4A was almost 4 times of KDM6A in the human liver cells (Supplementary Fig. 4d). Indeed, HNF4A is involved in the regulation of much broader physiological functions, such as lipid and glucose metabolism, hepatocyte development and proliferation[51]. HNF4A interacts with various coregulators, histone modifying complexes and TFs to activate the chromatin epigenetics and pioneer the accessibility of the chromatin[52,53]. For example, recent studies showed that HNF4A interplays with Histone Deacetylase 3 (HDAC3) and Prospero Homeobox Protein 1 (encoded by *PROX1*) to regulate liver triglyceride metabolic gene expression[54]. HNF4A is also required to direct MLL4 complex to maintain the H3K4me1 and enhancer latency in the liver cells[55], consistent with our finding that knockdown of HNF4A results in a global decrease of KDM6A binding, as well as in a reduction of H3K4me1[55]. Whether the HNF4A/MLL4 interaction requires KDM6A as an adaptor is unknown. KDM6A was reported to interact with pioneer factors in other cell and disease contexts. For example, in pancreatic acinar cells, HNF1A (an HNF family pioneer factor in the pancreatic acinar cells) recruits KDM6A to the activate differentiation programs to suppress pancreatic cancer[25]. HNF1A is predominantly expressed in the liver, similar regulatory mechanisms may also apply in the liver cells which can explain why KDM6A peaks were not totally abrogated upon HNF4A knockdown in our current study. Moreover, because HNF4A is an important Maturity Onset Diabetes of the Young (MODY) gene with multiple splicing variants, an important remaining question is whether disease related HNF4A variants differ in their affinity to KDM6A and thereby control epigenetics related with cholesterol metabolism in the liver[52,56,57].

The TF network analysis in HRO but not Huh1 cells has identified several common signal responsive TFs enriched in the KDM6A and HNF4A coregulated genes. Some of them such as Farnesoid X Receptor (FXR, also NR1H4) and CREBH (or CREB3L3) are well established regulators of liver lipid and cholesterol metabolism. Similar analysis in the LKO mice showed identical top TFs. However, most of FXR target genes were not responsive upon agonist treatment in the HRO cells. In the LKO mice, *Shp* (encoded by gene *Nr0b2*) was downregulated upon KDM6A knockout in the liver after GSK2324 (FXR agonist) treatment. SHP (also NR0B2) is an atypical orphan nuclear receptor and corepressor involved in suppressing bile acid synthesis genes, its downregulation therefore could not explain the accumulated cholesterol in the LKO liver, as well as reduced expression of key bile acid synthesis genes such as *Cyp7a1* and *Cyp8b1* in the AAV-PCSK9 mouse livers. *Bsep* was one of the very few FXR target genes[58] affected by KDM6A knockout after GSK2324 treatment. BSEP is a bile salt export pump and its decreased expression in the liver may lead to reduced bile acid export and accumulation of bile acids and cholesterol in the liver[58]. However, the effect on *BSEP* expression was minimal and unlikely to contribute significantly to the observed cholesterol accumulation. We believed that CREBH was the dominant TF affected by KDM6A depletion in the liver. Knockdown of CREBH decreased key lipoprotein and cholesterol metabolic genes such as *APOA1*, *APOM* and *APOH*, similar

to what was observed upon KDM6A or HNF4A knockdown in the HRO cells. In mice, both *Crebh* and *Kdm6a* LKO showed increased VLDL cholesterol levels and accelerated progression of atherosclerosis in the *Ldlr* deleted background. While in *Crebh* KO mice, the VLDL triglyceride was also increased while the *Kdm6a* LKO mice showed slight reduction in VLDL triglyceride[59]. Moreover, liver-specific overexpression of *Crebh* in the mice reduced both VLDL triglyceride and cholesterol, further confirming the phenotype[60]. It was proposed that *Apoa4* was the main regulator involved in VLDL regulation in the *Crebh* KO mice. As Apoa4 coactivates lipid lipase, its downregulation reduces lipid lipase activities, leading to accumulated total VLDL particles in the blood[59]. While in *Kdm6a* LKO mice, the VLDL production but not clearance was involved in the VLDL changes. It was also proposed that the triglyceride and cholesterol lowering effects of Crebh was through Apoe, as *Crebh* overexpression in the liver significantly induced *Apoe* expression and the protective role of Crebh was gone in *Apoe* KO mice[60]. Neither *Apoa4* nor *Apoe* expression levels were changed in *Kdm6a* LKO mice. These data suggested that although CREBH and KDM6A shared common regulatory genes, their transcriptomic target signatures are not entirely the same and therefore the knockout of both proteins in the liver may possess specific regulatory functions.

We also performed CREBH ChIP-seq upon KDM6A knockdown in the HRO cells. Despite the relatively low quality of the ChIP-seq results due to lack of highly competent ChIP grade antibodies, we could still observe decreased CREBH binding at *APOA1* and *APOC2/APOC4* cluster in the HRO cells. We have also successfully performed ChIP-seq in AAV-PCSK9 WT and LKO female mice. The results proved direct Crebh dynamics in many Kdm6a regulated lipoprotein and cholesterol metabolic genes including *Cyp7a1, Cyp8b1*, etc, further supporting the species conserved mechanisms underlying KDM6A regulation in the liver.

KDM6A shares high similarity in the enzymatic catalytic domain as KDM6B, therefore the inhibitor GSK-J4 hits both enzymes[61]. Multiple studies have shown that KDM6B deletion protects the hepatocytes from steatosis and lipotoxicity[62-64]. It is yet unclear whether KDM6B plays an important role in cholesterol metabolism. Our *Kdm6a* LKO mice had lowered liver triglyceride, suggesting common regulatory functions of KDM6A and KDM6B in liver lipid metabolism. Noteworthy, the KDM6A/B enzymatic inhibitor GSK-J4 seems to have similar functions[63,64], suggesting that such lipid lowering effects might depend on the demethylase activities of KDM6, in contrast to the catalytic-independent cholesterol modulating mechanisms of KDM6A in the liver. Therefore, it is reasonable to hypothesize that raising KDM6A levels in the liver may lower blood cholesterol but also possess the risk of inducing steatosis in the liver. On the contrary, it may also be hypothesized that KDM6A/6B dual enzymatic inhibitors may protect the liver from lipotoxicity without increasing atherogenic lipoprotein profiles. Further efforts are required to dissect the KDM6A/B demethylase-dependent and -independent networks of genes and pathways which could be separated by more specific activators or inhibitors.

We acknowledge several major limitations in the current study. Firstly, most experiments were conducted using cancer cell lines, which possess abnormal chromosome numbers and may not fully recapitulate normal human liver physiology. Future investigations should leverage more physiologically relevant models, such as primary human hepatocytes or liver organoids, to provide human-relevant evidence supporting the regulatory function of KDM6A in the human liver. Secondly, the phenotypic analysis relies heavily on observed mRNA changes. Although we validated several key enzymes (e.g., CYP7A1, DGAT2) via Western blot and confirmed multiple transcriptomic signaling pathways through LKO phenotypes, the study requires more evidence to mechanistically link KDM6A to the dysregulation of liver cholesterol and lipoprotein phenotypes. Finally, while

we successfully established the KDM6A-HNF4A-CREBH regulatory network in controlling cholesterol and lipoprotein metabolic genes within HRO human cell lines, we were unable to fully replicate the entire experimental set in mice. This was primarily due to the lack of high-quality, ChIP-grade antibodies for KDM6A in mouse liver tissue. However, the consistency observed in signaling pathway profiling, transcription factor enrichment analysis, and CREBH ChIP-seq results across both human cell and mouse models strongly suggests that the KDM6A-HNF4A-CREBH network is species-conserved in liver metabolism.

## Methods

### Mice

*Kdm6a*$^{flox/flox}$ mice were developed in Cyagen using a targeting construct which contains loxp sites flanking exon 5 of *Kdm6a*. To create the LKO mice, the *Kdm6a*$^{flox/flox}$ mice were crossed with *Alb*-Cre mice (B6.Cg-*Speer6-ps1*$^{Tg(Alb-cre)21Mgn}$/J) obtained from Jackson Laboratory (stock no. 003574). Both *Kdm6a*$^{flox/flox}$ and the *Alb*-Cre mice were bred with wild type C57BJ6 mice for at least 9 generations before breeding. The paired *Kdm6a*$^{flox/flox}$*Alb*-Cre$^{-/-}$ mice were used as negative controls. The mice are kept in IVC-cages with a house, nesting materials and gnawing stick. The cages used are Tecniplast GM500 (Greenline): 391 × 199 × 160 (172*) mm (501 cm$^2$) at PKL4 and Allentown typ II Long V12,: 213 × 362 × 175 (127) mm (542 cm$^2$) or Allentown NexGen: 194 × 178 × 397 (535 cm$^2$) at PKL3 and 5. Max 5 animals in one cage. The temperature are 22 ± 2 degrees and the humidity 50 ± 5. The light cycle is 12/12.

**Atherosclerosis model.** To generate the atherosclerosis mouse models, twelve-week-old control (WT) and LKO mice received a single intraperitoneal injection of 10$^{11}$ particles of rAAV8/D377Y-mPCSK9 per mouse. The mice were then fed a western diet which contains 40% fat, 0.5% cholesterol (D12107C, Research Diets) for 12 weeks. Body weight and blood glucose levels were measured at specified time points using an electronic weighing scale and a glucometer (Accu-Chek Performa, Roche).

Atherosclerosis quantification[35] was performed at the end of the treatment period, mice were euthanized using $CO_2$ asphyxiation followed by cervical dislocation. The aorta was carefully dissected from the aortic root to the iliac bifurcation, and surrounding adipose tissue was removed. The aortic root was embedded in optimum cutting temperature (OCT) compound and stored at -80 °C until cryosectioning. Sections were stained with Oil Red O to visualize lipid deposits and counterstained with hematoxylin. Stained sections were examined under a light microscope, and images were captured using a digital camera. The extent of atherosclerotic lesions was quantified using ImageJ software.

**Lipid and cholesterol analysis.** Plasma lipoproteins were fractionated from 2.5 µl of plasma samples from each mouse using a Superose 6 INCREASE 3.2/30 column (GE Healthcare), followed by online determination of triglycerides and cholesterol levels[65]. Liver TG concentration was determined using a Triglyceride Quantification Kit (MAK266, Merck), and liver cholesterol concentration was determined using a Cholesterol Quantification Assay Kit (CS0005, Merck) according to the protocols. Liver, serum and feces total bile acid concentration was determined using a Mouse Total Bile Acids Assay Kit (80471, Crystal Chem) according to the protocol. Mouse plasma APOB100 was measured using an ELISA kit (ab230932, abcam) according to the protocol. C4 and bile acids in serum were analyzed by LC-MS/MS using deuterium-labeled standards of unconjugated and taurine-conjugated BAs using protocols described previously[66,67]. Deuterium-labeled standards of unconjugated and glycine- and taurine-conjugated BAs were from Steraloids, Inc. (Newport, RI).

**Bile acids and C4 analysis.** Individual bile acids in serum were quantified by liquid chromatography–tandem mass spectrometry (LC-MS/MS) using deuterium-labelled internal standards for unconjugated, glycine- and taurine-conjugated bile acids, their sulfate metabolites, and the bile acid synthesis marker C4, as previously described. Deuterium-labelled standards were obtained from Steraloids, Inc. (Newport, RI, USA). Data were acquired on a Xevo TQ-XS triple quadrupole mass spectrometer (Waters Corp., MA, USA) coupled to an Acquity UPLC system and processed using MassLynx software (Waters Corp., MA, USA). A total of 10 serum samples from mice were analyzed, comprising 2 experimental groups with 5 mice in each group. Each biological sample was prepared and injected once (n = 1 technical replicate), alongside pooled QC samples and blank injections to assess carry-over and analytical variability. 50 µL serum were spiked with 10 µL of each deuterium-labelled C4 and bile acid internal standard solution (10 ng d7-C4 and 25 ng d4-bile acid), mixed with 400 µL acetonitrile and vortexed for 5 s. After centrifugation for 15 min at 14 000 g, supernatants were transferred to glass tubes and evaporated to dryness under a stream of nitrogen at 40 °C. Residues were reconstituted in 200 µL methanol–water (1:1, v/v), and 5 µL were injected for LC–MS/MS analysis. HPLC-grade water, methanol (LC–MS grade), acetonitrile, and isopropanol were used together with formic acid as modifiers. Chromatography was performed on an Acquity UPLC BEH C18 column (1.7 µm, 2.1 × 50 mm; Waters Corp., MA, USA) maintained at 65 °C with a flow rate of 0.5 mL/min and an injection volume of 5 µL. The autosampler was kept at 10 °C. Mobile phase A consisted of water with 0.1% formic acid, and mobile phase B consisted of acetonitrile with 0.1% formic acid. The gradient (total run time 22 min) started at 95% A/5% B, was held to 0.5 min, then changed to 75% A/25% B at 5.0 min, 60% A/40% B at 10.5 min, and 5% A/95% B at 17.5–19.0 min, before returning to 95% A/5% B at 19.5–22.0 min for column re-equilibration. Weak and strong needle wash solutions were 10% methanol in water and isopropanol:methanol:acetonitrile:water (25:25:25:25, v/v/v/v), respectively. Analyses were performed on a Xevo TQ-XS triple quadrupole mass spectrometer (Waters Corp., MA, USA) equipped with an electrospray ionization source operated in positive ion mode for C4 and in negative ion mode for bile acids. The vaporizer temperature was 350 °C, spray voltage 3.5 kV, desolvation gas flow 1000 L/h, cone gas 150 L/h, and nebuliser gas 7 bar. Data were acquired in multiple reaction monitoring (MRM) mode using optimized precursor/product ion transitions and compound-specific cone voltages and collision energies for each analyte and its corresponding deuterium-labelled internal standard. The following bile acids and related compounds were quantified together with their deuterium-labelled internal standards: C4/C4-d7, CA/CA-d4, CDCA/CDCA-d4, DCA/DCA-d4, GCA/GCA-d4, GCDCA/GCDCA-d4, GDCA/GDCA-d4, GLCA/GLCA-d4, GUDCA/GUDCA-d4, LCA/LCA-d4, TCA/TCA-d4, TCDCA/TCDCA-d4, TDCA/TDCA-d4, TLCA/TLCA-d4, TUDCA/TUDCA-d4, UDCA/UDCA-d4, α-MCA/α-MCA-d4, β-MCA/β-MCA-d4, ω-MCA/ω-MCA-d4, Tα-MCA/Tα-MCA-d4, Tβ-MCA/Tβ-MCA-d4, and Tω-MCA/Tω-MCA-d4.

Peak integration and quantification were performed in MassLynx, using calibration curves constructed from authentic standards with internal-standard normalization. Analyte identity was confirmed by matching retention times and MRM transitions to standards; QC samples were used to monitor precision and stability over the batch.

**VLDL production assay.** Hepatic VLDL secretion assay was performed before the end of the treatment period, mice were fasted for 5 hours prior to the experiment. Following the fasting period, baseline blood (~75 µL) was collected. Immediately afterwards, mice received an intraperitoneal injection of poloxamer-407 (16758, Sigma) at a dose of 1.0 g/kg body weight. The P-407 solution was prepared as a 10% (w/v) suspension in saline. Subsequent blood samples (~75 µL each) were

collected at 1, 2, and 5 hours after P-407 administration. Plasma lipoprotein levels were measured using FPLC as mentioned above.

**VLDL isolation.** Plasma samples (100 μL each) were supplemented with 900 μL of KBr solution (1.019 g/mL) and their densities were adjusted to 1.019 g/mL using KBr. The samples were transferred to 1 mL thick-wall Polycarbonate tubes (Backman Coulter, USA). The tubes were ultracentrifuged for 20 h at 244000xg at 4 °C (Optima MAX-TL Ultracentrifuge, rotor TLA 120.2, Beckman Coulter, USA). Postcentrifugation, the top VLDL-rich fractions were transferred into the new tubes for further refinement. For the second ultracentrifugation step, the VLDL-rich fractions were further brought to the volume of 1 mL using KBr solution (1.019 g/mL) and adjusted to 1.019 g/mL. The ultracentrifugation step was repeated and top VLDL-rich fractions collected. These fractions were further concentrated using Amicon Ultra centrifugal concentrators (0.5 mL, 3 kDa cut-off). This step was repeated twice to replace KBr with PBS. The size of particles present in the VLDL fractions was assessed by Dynamic Light Scattering using Zetasizer (Malvern). Furthermore, 20 μL of each VLDL fraction were transferred to the epi and labelled with 0.5 μL of NR12S[37] (0.1 mg/mL in DMSO).

**Single Particle Profiling.** Single Particle Profiling (SPP) was performed using the setup for fluorescence correlation spectroscopy on a Zeiss LSM 780 microscope[36]. A 488 nm argon ion laser was used for NR12S excitation[37]. A 40 × 1.2 NA water immersion objective was used to focus the light. The laser power was set to 1% of the total laser power that corresponds to 20 μW. The emission detection windows on the single photon counting GaAsP detector were set to 490–560 for green and 650–700 for red parts of the spectrum. Emission from NR12S was recorded simultaneously in both channels. Forty intensity traces of 15 s were recorded. Traces were then analysed using the python-based software "Py Profiler".

**Cell culture**

Huh7 (RRID: CVCL_0336, Cytion-300156), SNU-182 (RRID: CVCL_0090, Cytion-305119), Huh-1 (RRID: CVCL_2956, JCRB- JCRB0199, established by Hoh,H.) and HROHep03 (RRID: CVCL_2U72, Cytion-300197) cell lines were tested for mycoplasma contamination before the experiments. Huh7 and Huh-1 cells were cultured in Dulbecco's Modified Eagle Medium (DMEM, D5648, Merck) supplemented with 10% fetal bovine serum (FBS, SH30071.03, HyClone) and 1% Penicillin-Streptomycin (P4458, Merck). SNU-182 cells were cultured in RPMI 1640 (A10491-01, Gibco) supplemented with 10% fetal bovine serum (FBS, SH30071.03, HyClone) and 1% Penicillin-Streptomycin (P4458, Merck). HROHep03 cells were maintained in high-glucose DMEM/F12 medium (SLM-243-B, Merck), supplemented with 5% FBS (SH30071.03, HyClone) and 1% Penicillin-Streptomycin (P4458, Merck). Both cell lines were incubated at 37 °C in an atmosphere of 5% CO2.

**Karyotyping**

Genomic DNA was extracted from the cell lines HROhep03, Huh-1, Huh7, SNU-182, and HepG2 using the Blood & Cell Culture DNA Mini Kit (13323, Qiagen). Male primary human aortic smooth muscle cells (ATCC, PCS-100-012) were used as a diploid reference control. Quantitative PCR (qPCR) was performed with equal amounts of DNA from each cell line to target the Amelogenin X-Linked (*AMELX*) gene (Forward: 5' GCTTGCCTCTGCTGAAATATTAGTG3', Reverse: 5'CTCATG-CATTCCGCTGTTCTG3'. The copy number of the X chromosome in each cell line was calculated relative to the HASMC control using the formula: X chromosome copy number = $2^{\wedge}(\Delta CT)$, where $\Delta CT = CT$ (PASMC) - CT (cell line).

**siRNA transfection**

Cells were seeded in 6-well plates at a density of $7.5 \times 10^5$ cells/well and incubated for 12 h in their respective culture media to achieve 60–80%

confluency at the time of transfection. siRNA targeting KDM6A, HNF4A, KMT2C, KMT2D and CREBH, as well as negative control siRNA, 10 nmol siRNA was used for each well. Transfections were performed using Lipofectamine RNAiMAX Transfection Reagent (ThermoFisher) following the manufacturer's protocol. Briefly, Lipofectamine RNAi-MAX Reagent and siRNA were each diluted in Opti-MEM® Medium and then combined at a 1:1 ratio. The siRNA-lipid complexes were allowed to form by incubating the mixture for 5 minutes at room temperature. The complexes were then added to the cells, and the transfection was carried out for 48 h at 37 °C. Subsequent experiments were performed following the transfection period. The siRNA sequences were listed in Supplementary Table 1.

**qPCR analysis**

Total RNA was extracted from snap-frozen liver tissues and human cell lines using the E.Z.N.A.® Total RNA Kit I (R6834-02, Omega Bio-tek). Complementary DNA (cDNA) synthesis was performed using M-MLV Reverse Transcriptase (28025-021, Life Technologies). qPCR was conducted on the ABI Prism 7500 PCR system (Applied Biosystems). For normalization, *36b4* (acidic ribosomal phosphoprotein P0, also known as *RplpO*) was used as the house keeping gene in mouse liver, while *Ubiquitin C* (*UBC*) and *Hydroxymethylbilane Synthase* (*HMBS*) were used as house keeping genes for human cell lines. Relative changes in mRNA expression were calculated using the comparative cycle threshold method ($2^{-\Delta\Delta Ct}$). Primer sequences are listed in Supplementary Table 2.

**RNA-seq**

RNA was extracted from mouse liver biopsies or human cell lines using the kit stated above. RNA quality was assessed using the 5400 Fragment Analyzer System (Agilent). mRNA library preparation was carried out using the Novogene NGS RNA Library Prep Set (PT042, Novovene) following the manufacturer's protocol. RNA-seq libraries were sequenced on the NovaSeq X Plus Series (PE150) platform at NOVO-GENE (Cambridge, United Kingdom). Preprocessed reads were aligned to the NCBI38/mm10 (for mouse) or GRCh38/hg38 (for human) genomes using the HISAT2 program, and read counts were determined using featureCounts v1.5.0-p3[68]. Raw tag counts were imported into R and Bioconductor, and differential gene expression was analyzed using the DESeq2 package. Transcription factor enrichment analysis was carried out using CHEA3[41].

**Western-blotting**

Protein extraction from liver tissues was carried out with RIPA Lysis Buffer containing phenylmethylsulfonyl fluoride, protease inhibitor, and phosphatase inhibitor. The protein concentration was determined using a bicinchoninic acid (BCA) assay kit (71285-M,Merck). Subsequently, the samples were subjected to electrophoresis on 8% SDS-polyacrylamide gels and transferred onto nitrocellulose membranes. The PVDF membranes were blocked with 5% skimmed milk for 2 h, followed by an overnight incubation at 4 °C with specific primary antibodies. The primary antibodies targeted the following proteins: SREBF1 (14088-1-AP, Proteintech), LCAT (12243-1-AP, Proteintech), DGAT2 (17100-1-AP, Proteintech), CYP7A1 (ab65596, Abcam) and -β-actin (ab8226, Abcam). The next day, the membranes were incubated with peroxidase conjugated goat anti-mouse IgG secondary antibodies (AP124P, Merck) or peroxidase conjugated donkey anti-rabbit IgG secondary antibodies (NA934, Cytiva). Protein bands were visualized using a Clarity Western ECL Substrate kit (170-5061, BIO-RAD).

**ChIP-seq and RIME**

Fresh liver tissues (chopped into small pieces) and HRO or Huh1 cells were crosslinked with 1% formaldehyde (28906, ThermoFisher) in PBS for 10 minutes for histone modifications. For TFs or coregulators, tissues/cells were double crosslinked with 2 mM disuccinimidyl glutarate

(DSG) (20593, ThermoFisher) for 30 minutes, followed by 1% formaldehyde for 10 minutes. The reaction was stopped with glycine at a final concentration of 0.125 M for 5 minutes. Liver pieces were disaggregated in ice-cold PBS with protease inhibitor (12352204, Roche) using a dounce homogenizer, first with a loose pestle and then with a tight pestle (FB56691, Fisher Science). Nuclei were isolated using three lysis buffers: lysis buffer 1 (50 mM Hepes–KOH, pH 7.5, 140 mM NaCl, 1 mM EDTA, 10% glycerol, 0.5% IGEPAL CA-630, and 0.25% Triton X-100), lysis buffer 2 (10 mM Tris–HCl, pH 8.0, 200 mM NaCl, 1 mM EDTA, and 0.5 mM EGTA), and lysis buffer 3 (10 mM Tris–HCl, pH 8.0, 100 mM NaCl, 1 mM EDTA, 0.5 mM EGTA, 0.1% Na-deoxycholate, and 0.5% N-Lauroylsarcosine). The samples were then sonicated for 30 minutes (30 seconds ON/30 seconds OFF) using the Bioruptor Pico (Diagenode). Protein A Dynabeads (10001D, Invitrogen) were incubated overnight with the antibodies. Each lysate was immunoprecipitated with the following antibodies: control rabbit IgG (sc-2027, Santa Cruz, 1–5 μg), anti-H3K27ac (ab177178, Abcam, 1 μg), anti-H3K4me1 (ab8895, Abcam, 1 μg), anti-H3K4me2 (ab7766, Abcam, 1 μg), anti-H3K27me3 ((07-449, Millipore, 1 μg), and anti-CREBH (HPA040671, Sigma Aldrich, 1 μg). The immunoprecipitated Protein A beads were washed 3 times with lysis buffer and sent for proteomics analysis (RIME[39]).

## The samples for proteomics were subjected to SDS-PAGE

Coomassie-stained bands were manually excised from the gel. The in-gel digestion was performed. Samples were suspended in 0.1% TFA and analyzed using an Ultimate 3000 liquid chromatography system coupled to an Orbitrap QE HF (Thermo Fisher). Briefly, peptides were separated in a 30 min linear gradient starting from 3% B solution (0.1% formic acid, 89.9% acetonitrile) and increasing to 23% B over 25 min and to 38% B over 5 min, followed by washout with 95% B. The mass spectrometer was operated in data-dependent acquisition mode, automatically switching between MS and MS2. MS spectra (m/z 400–1600) were acquired in the Orbitrap at 60,000 (m/z 400) resolution and MS2 spectra were generated for up to 15 precursors with a normalized collision energy of 27 and an isolation width of 1.4 m/z. The MS/MS spectra were searched against the UniProt H. sapiens (UP000005640, downloaded on June 2020), and a customized contaminant database using Proteome Discoverer 2.5 with Sequest HT. The fragment ion mass tolerance was set to 0.02 Da and the parent ion mass tolerance to 5 ppm. Trypsin was specified as an enzyme. The following variable modifications were allowed: Oxidation (M), Deamidation (N, Q), Acetylation (N-terminus), Met-loss (M), and a combination of Met-loss and acetylation (N-terminus), whereas Carbamidomethylation (C) was set as a fixed modification. Peptide quantification was done using a precursor ion quantifier node with the Summed Intensity method set for protein abundance calculation.

## For ChIP-seq library preparation, the same ChIP protocol was followed

Formaldehyde cross-linking was reversed overnight at 65 °C. The ChIP DNA was purified using the ChIP DNA Clean and Concentrator Capped Zymo-Spin I (Zymo Research) purification kit. Two to four ChIPs were pooled during the final purification step to obtain concentrated material. For library preparation and sequencing, 2–10 ng of ChIPed DNA was processed using the Rubicon ThruPLEX DNA-seq kit (TAKARA) following standard protocols, and sequenced on the NovaSeq X Plus Series (150PE reads, Novogene).

## ChIP-seq data analysis

Computations of mice data were performed using resources provided by Galaxy[69]. The computations of human data was enabled by resources in project sens2024555 provided by the National Academic Infrastructure for Supercomputing in Sweden (NAISS) at UPPMAX.

Analysis was conducted as previously described[65]. Sequencing files (fastq) provided by Novogene (Cambridge, United Kingdom) or BEA core facility (Stockholm, Karolinska Institute), along with the raw data from published ChIP-seq datasets (KDM6A: GSE95890; HNF4A: GSE31477)[40], were aligned to the NCBI38/mm10 version of the mouse reference genome or GRCh38/hg38 using Bowtie2. Peaks were identified using the HOMER package.

Peak heights were normalized to the total number of uniquely mapped reads and displayed in the Integrative Genomics Viewer (IGV) as the number of tags per 10 million tags. For statistical analysis of the peaks, raw tag counts were imported into R and Bioconductor, and the edgeR package was used to identify potential differential-binding sites.

## CUT&Tag sample preparation

CUT&Tag sample preparation was performed following published protocols from the Henikoff lab[70]. Approximately 500,000 HRO or Huh1 cells were harvested and counted and were resuspended in Wash Buffer and mixed with Concanavalin A-coated magnetic beads. The mixture was incubated at room temperature to allow binding. Bead-bound cells were then washed and incubated with the anti-KDM6A (PA5-31828, Invitrogen, 1 μg) primary antibodies specific to the target protein, followed by a secondary antibody (PA5-31828, EpiCypher) for bridging. Next, the bead-bound cells were incubated with pAG-Tn5 (79561, EpiCypher) in the presence of digitonin for tethering. After a brief incubation to allow the transposome to tether to chromatin sites targeted by antibodies, cells were washed and subjected to magnesium activation to induce tagmentation. This step facilitated DNA cleavage and simultaneous tagging with sequencing adapters. Following tagmentation, DNA was extracted and quantified using Qubit fluorometer (Thermo Fisher) following the instructions. Libraries were PCR-amplified with barcoded primers, cleaned up using SPRI beads, and subjected to quality control checks, including size selection and quantification. The prepared libraries were sequenced on the NovaSeq X Plus Series (150PE reads, Novogene, United Kingdom).

## CUT&Tag data analysis

Sequencing data was processed using standard pipelines. Sequencing files (fastq) provided by Novogene (Cambridge, United Kingdom) were aligned to the GRCh38/hg38 version of the human reference genome using Bowtie2. Duplicate reads were removed, and peaks were called using the HOMER package to identify regions of significant enrichment. SEACR[71] is used to call peaks and enriched regions from chromatin profiling data. ChIPseqSpikeInFree[72] is used to normalize Data.

## FXR agonist treatment in mice and human cell line

Twelve-week-old WT and LKO mice were administered either a vehicle (H$_2$O) or GSK2324 (dissolved in H$_2$O) via intraperitoneal injection at a dosage of 30 mg/kg body weight once daily for three days. The animals were allowed to feed ad libitum until the final day of treatment. On this day, GSK2324 (Sigma Aldrich) was administered starting at 9 am, followed by a four-h fasting period. The mice were then sacrificed starting at 1 pm to maintain a consistent circadian cycle. For the cell experiments, total RNA was extracted from the human cell line HROhep03 following a 24-h treatment with 1 μM GW4064. Subsequent experiments were then conducted using the isolated RNA.

## LXR agonist treatment in mice hepatocytes and human cell line

Twelve-week-old WT and LKO mice were sacrificed. GentleMACS was used to perfuse livers, dissociate the tissues, and isolate hepatocytes. Total RNA was extracted from the hepatocytes following a 6-h treatment with 1 μM GW3965/T0901317 (Sigma Aldrich). For human cell line, total RNA was extracted from the hepatocytes following a 24-h treatment with 1 μM T0901317.

**Rare variant gene-level associations for *KDM6A* with serum ApoA1 and ApoB levels**

The data was derived from the Type 2 Diabetes Knowledge Portal website. Passing Variants indicate the number of variants in the analysis, which were grouped into sets using two criteria: variants predicted with high confidence by Loss-Of-Function Transcript Effect Estimator (LofTee), or missense variants as well as variants predicted with low confidence by Loss-Of-Function Transcript Effect Estimator (LofTee).

**Statistical analysis**

All the RNA-seq, ChIP-seq, CUT&Tag, qPCR experiments and the phenotype analysis in cell and mouse-based studies were conducted with biological replicates, with minimum $n = 3$ in each group. The number of replicates and the statistical analysis in each experiment are described in the figure legends. Sample sizes were not predetermined statistically. The D'Agostino and Pearson normality test was employed to assess normal distribution. Variance within each data group was compared using the F-test for two groups or the Brown–Forsythe test for more than two groups. Statistical analyses were performed using GraphPad Prism 10 (GraphPad Software, Inc., La Jolla, CA), with data represented as mean ± standard error of the mean (s.e.m.). Statistical tests were conducted after confirming that the data met the appropriate assumptions of normality, homogenous variance, and independent sampling. All tests were two-tailed, with significance defined as $p < 0.05$. No statistical methods were used to predetermine sample size. Grubbs' test was performed to identify an outlier. In general, all samples and animals were included in the analysis.

**Ethics**

All animal experiments were approved by Swedish Board of Agriculture with ethic permit number 05517-2022 and ID907. The experiments were conducted in accordance with the International Guiding Principles for Biomedical Research Involving Animals, as developed by the Council for International Organizations of Medical Sciences (CIOMS). The mice were bred and maintained at the Center for Comparative Medicine at Karolinska Institutet and University Hospital (PKL, Huddinge, Sweden).

**Reporting summary**

Further information on research design is available in the Nature Portfolio Reporting Summary linked to this article.

## Data availability

Gene expression RNA-seq data, CUT&Tag and ChIP-seq data have been deposited at the NCBI Gene Expression Omnibus (GEO) accession numbers are GSE287688, GSE287680 and GSE287736. The mass spectrometry proteomics data for RIME have been deposited to the ProteomeXchange Consortium via the PRIDE partner repository with the dataset identifier PXD074624. Other data are available from the corresponding author upon request. Source data are provided with this paper.

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

## Acknowledgements

R.F. was supported by grants from EFSD Novo Nordisk future leaders award, Swedish Research Council (2023-02311), Swedish Cancer Society (232891 Pj), the Karolinska Center for Innovative Medicine CIMED (FoUI-975445) and Karolinska Institute Strategic Research Programme in Diabetes (SRP) Rolf Luft Grant. E.T. was supported by Horizin EU INTERCEPT-T2D (101095433). B.A was supported by Swedish Research Council, Swedish Heart and Lung Foundation. L.C and Q.L were supported by studentship from Chinese Scholarship Council (CSC). Z.F.K was supported by the Guangdong Basic and Applied Basic Research Foundation (2026A1515010674). We acknowledge Matteo Titus, and Marina Stoimenou for the preliminary experiments and bioinformatics analysis to initiate this study. We received the technical support of Core Facility for Mass Spectrometry and Proteomics (CFMP, DFG RI_00574) of the Center for Molecular Biology (ZMBH) of Heidelberg University. We thank Marcin Luzarowski for support with mass spectrometry analysis. Core Facility for Mass Spectrometry and Proteomics is funded by the ZMBH and partially funded by the CellNetworks Core Technology Platform (CCTP) of Heidelberg University. The CCTP is funded in part by the Federal Ministry of Education and Research (BMBF) and the Ministry of Science Baden Württemberg within the framework of the Excellence Strategy of the Federal and State Governments of Germany. We also thank the proteomics support from Clinical Proteomics Mass Spectrometry Facility (CPMSF) at Karolinska Institutet. We thank Professor Uwe Tietge (Karolinska Institutet) for the helpful discussion and input for this study. We also thank Haonan Li for providing the protocol for the histology analysis and Dr. Marianna Skipitari from Maria Eriksson group for providing the HASMC cells.

## Author contributions

R.R.F. designed the study, provided the fundings, performed part of the cell animal experiments and analyzed all the data in the manuscript, coordinated the project, discussed with other coauthors and wrote the manuscript; Z.F.K. initiated the study together with R.R.F., purchased the knockout animals, designed the animal studies and performed the preliminary animal experiments; L.C. performed most of the cell and animal experiments, analyzed the data and participated in drafting the manuscript under the supervision of R.R.F.; Z.Y.L., R.N.L., S.S., Z.Q.H. and P.V.A. did part of the cell experiments and analyzed the data; O.G.I., R.N.L., P.B.L. and X.R.Z. participated in developing the atherosclerosis animal model and contributed to the pathological analysis of the aortic roots; J.H., M.P., P.P. and B.A. supported the FPLC lipoprotein faction analysis and provided suggestions to the study; J.H. did the LC-MS/MS analysis of bile acid and C4; T.S. and E.S. analyzed the VLDL particles using P.P.; Q.L. and A.B. analyzed the public human datasets; R.R.F., K.Z.F. and J.W.Z. generated the LKO mice; E.T. provided technical and intellectual support to the study and also participated in the manuscript drafting.

## Funding

## Competing interests

The authors declare no competing interest.
