## [Transparent Peer Review file · Nature Communications]

Sex-specific KDM6A-HNF4A-CREBH network controls lipoprotein cholesterol metabolism and atherosclerosis via epigenetic reprogramming of hepatocytes

Corresponding Author: Dr Rongrong Fan

Version 0:

Reviewer comments:

Reviewer #2

(Remarks to the Author)

The manuscript studies an underrepresented area – the molecular basis beyond the sexual dimorphism of the liver. The rationale of the study is a known phenomenon that the expression of KDM6A (Lysine Demethylase 6A) differs between females and males and that its expression can be lost in liver cancers. KDM6A is located on X chromosome and since it can escape X inactivation it is expressed higher in female compared to the male tissues. By integrating transcriptomic and epigenomic techniques in human liver cell lines and liver-specific knockout mice and disease models, the authors aimed to investigate the physiological roles and regulatory mechanisms of the hepatic KDM6A. They promise to answer the question what is the metabolic function of KDM6A in regulating liver lipid metabolism, how such regulation is modulated in the chromatin epigenetic levels and whether liver KDM6A deficiency leads to sex difference in developing metabolic disorders. The questions are relevant and novel but I am not sure that the answers provided are always clear and sufficiently general. I question also whether the sex dimorphism has been proven for the KDM6A-HNF4A-CREBH network.

To draw the conclusions the authors used two hepatic cell lines of different sexes (female line HROHep03 and male cell line Huh7, both from Cytion) HROHep03 is a primary liver adenocarcinoma cell line; Huh7 is a well-differentiated hepatocellular carcinoma (HCC) cell line. They also used mouse strains, Kdm6flox/flox developed by Cyagen and WT controls, with or without the mPCSK9 delivered by adenovirus.

Major remarks.

Studies in cell lines: While both HROHep03 and Huh7 cell lines originate from liver cancers in individuals of different sexes, their cancerous nature and associated chromosomal abnormalities complicate the assessments of X-chromosome gene dosage. For precise determination, experimental analyses such as karyotyping, quantitative PCR, or RNA sequencing would be necessary to assess the X-chromosome content and gene expression levels in these cell lines. This would prove whether these two cell lines are good models to study KDM6A in relation to sex dimorphic differences in metabolic pathways, potentially leading to disorders. The authors should thus determine the X-chromosome and KDM6A gene dosage of both cell lines to illuminate some of their conclusions.

For example, the gene ontology analysis in KDM6A knockdown male and female liver cell lines showed similar cholesterol and lipoprotein modulating pathways, which is unexpected taking in consideration the envisioned higher expression of KDM6A in females.

For some later mechanistic studies it is not always clear why sometimes only HRO cell have been used while in other cases only Huh7. If any comparison is to be made one would expect parallel experiments in both cell lines, even if they are not optimal. Or at least a more structured representation of experiment and data flow, considering the sex dimorphism in the focus.

Studies in the mouse models: Authors showed that the atherogenic lipoprotein profiles were only observed in female but not male KDM6A LKO mice. Was the background of the KDM6A flox/flox mice 100% B6? Since for B6 the escape from X-inactivation has been documented, but I am not sure for other mouse lines.

Chip-Seq – why was the Chip-seq analysis for methylation and acetylation KDM6A activities performed only in the HRO cells? Is it true also for Huh7 cells (or male cells) that KDM6A regulation of lipoprotein metabolic genes is independent of H3K27me3 demethylation? I am not convinced that the Chip_seq data in the female floxed and LKO mouse liver data are confirmatory of the HRO cell data.

To identify the TFs responsible for KDM6A recruitment to the lipoprotein and cholesterol (LnC) genes - HNF4A knockdown reduced KDM6A peaks in both Huh7 and HRO cells, so based on the cell lines, this is a general and not sex-specific event.

Suppl. Fig 4 (j) shows the heatmap that compares the downregulated genes in KDM6A LKO female mice and HNF4A LKO male mice (GSE173968). This is not specified in the text of the main paper: "We also compared liver RNA-seq of the female LKO mice with HNF4A LKO mice (GSE173968)39, and found that around 30% of the downregulated genes in the KDM6A LKO mice were overlapping with HNF4A LKO". Claiming that the data was consistent with the HRO results does not prove that this is a sex dimorphic event.

KDM6A removal affects the binding and transcription activation of CREBH- the RNA-seq data from HRO and Huh7 cells identified several common TFs, where among the top candidates was also CREBH. Again, where is the sex dimorphism?

The title is over-ambitious since sex dimorphism was not experimentally proven for all components promised in the title. "Sex dimorphic KDM6A-HNF4A-CREBH network controls lipoprotein cholesterol metabolism and atherosclerosis via epigenetic reprogramming of hepatocytes"

The statement "We have therefore identified a novel sex-specific cholesterol regulatory mechanism in the liver." is over ambitious. The authors have to specify clearly what is indeed sexually dimorphic (and what common to both sexes), and based on which model the conclusions have been made. It is not a problem that the data from cell lines do not always correlate with the mouse data. The problem is "generalization" of findings in cases when the experiments do not support the case.

Minor comments:

- The background line of the Kdm6aflox/flox mice is not described.
- There is no data availability statement and location where transcriptome/ChIP-seq is deposited.
- Some smaller text corrections/typos on page 6 line 4,7, page 8 line 14, page 10 line 22 and page 12 line 14.
- In the Method part CUT&Tag sample preparation authors state "DNA was purified using SPRI beads and quantified using a fluorometric method". Detail on "fluorometric method" should be added. It should also be stated what is the origin of "50.000 harvested cells".

Reviewer #3

(Remarks to the Author)

In this study, the authors demonstrated that KDM6A regulates the expression of lipoprotein metabolism-related genes in a sex-specific manner. Using mice, they showed that hepatic loss of KDM6A results in elevated plasma cholesterol levels and accelerates the development of atherosclerosis in female mice but not in male mice. KDM6A is recruited to chromatin by HNF4A and, through CREBH, activates the transcription of lipid-related genes.

This is a very interesting study, clearly presented and well-written. However, I have some comments that could help improve this study.

In Figure 1, the authors showed that the loss of KDM6A leads to a decrease in mRNA levels encoding proteins involved in lipid and cholesterol metabolism, such as LIPC, HMGCR, and MTTP. This downregulation was conserved between human cells and mouse livers.

Interestingly, despite the downregulation of many lipid-related genes, plasma lipid levels increased. In the absence of LDLR, there was a two-fold increase in VLDL. How is this possible? As shown in Supplemental Figure 1i, genes involved in VLDL biogenesis were decreased. Do the authors have an explanation for this discrepancy? The study mainly focuses on gene expression, but did the authors also observe effects at the protein level? Is the increase in VLDL due to enhanced VLDL biogenesis, increased cholesterol synthesis, or a defect in clearance? Essential pathways that has to be studied to explain the mechanism underlying the lipid phenotype, as the lipid phenotype cannot be fully explained by the gene expression data. Similarly, for liver lipid levels, there is no clear correlation between the decrease in lipid-related genes and liver lipid accumulation. Is lipogenesis increased, or beta-oxidation?

Next, the authors showed that HNF4A is required for the recruitment of KDM6A to lipid gene loci. In Supplemental Figure 4j, they demonstrated an overlap in genes regulated by KDM6A and HNF4A in the mouse liver. What is the effect of HNF4A downregulation on plasma and liver lipids? Do these two models exhibit the same phenotype? Additionally, liver-specific CREBH-deficient mice share a similar phenotype with KDM6A LKO mice. The authors discuss the HDL phenotype but not the VLDL/LDL phenotype. Can they speculate on the underlying mechanism? The gene expression data do not seem to fully support the proposed explanation.

Minor points:

Page 8: "...while at adjacent CLPTM1 loci (?)" This appears to be an editing error that was not removed before submission.

Page 10: "KDM6A interplays [interacts?] with..."—this may be an editing error.

Page 12: "KDM6A in regulating (??):" This also seems to be a typo.

Reviewer #4

(Remarks to the Author)

Version 1:

Reviewer comments:

Reviewer #2

(Remarks to the Author)

The authors addressed Reviewer 2's concerns about sex-specific modeling by performing karyotype and X/Y chromosome analyses and identifying Huh1 as a suitable male human liver cell line for comparison with the female HRO cells. Huh1 cells were added because the previously used Huh7 line lacked the Y chromosome and thus could not accurately represent a male model; in contrast, Huh1 cells possess a normal male karyotype, including functional KDM6C/UTY expression. This is a crucial limitation when using cell lines to study sex differences. The authors performed also a new RNA-seq and ChIP-seq experiments in Huh1 cells, confirming that KDM6A regulation in males differs from females and is largely independent of its demethylase activity. The mechanistic evidence was expanded by analyzing HNF4A–KDM6A interactions and CREBH binding, demonstrating a sex-specific regulatory axis in lipid and cholesterol metabolism. The title has been modified to clarify the statements about sex dimorphism, methodological data and data-availability details have now been added. There are still some remaining inconsistencies or concerns.

1. Sex-dimorphism generalization remains overstated

Although the authors revised the title and toned down some claims, parts of the Discussion still generalize sex-specific conclusions. Many experiments proving "sex specificity" rely on only one female (HRO) and one male (Huh1) liver cell line, with Huh-7 remaining in some cases. Since all are cancer-derived and have abnormal karyotypes, it is not certain that the observed effects truly represent physiological male–female differences. The study still partially equates "cell line sex" with biological sex dimorphism, which remains an overextension.

I thus suggest "female" and "male" cell line to be replaced by "cell line derived from a female or from a male patient "

2. Remaining limited validation of HNF4A–KDM6A–CREBH axis in vivo.

The mechanistic link between KDM6A, HNF4A, and CREBH is well supported in cell models, but less robust in mouse experiments. For instance, ChIP-seq and CUT&Tag evidence in mice support co-regulation of a few target genes (e.g., ApoA1), but direct biochemical evidence of complex formation in vivo is not shown. This weakens the conclusion that the full "KDM6A–HNF4A–CREBH regulatory axis" is sex-specific and functional in the intact liver.

3. Statistical and replication transparency

Although the Methods section now provides more statistical details, biological replicates and batch numbers for some omics experiments (especially ChIP-seq and CUT&Tag) remain unclear. The paper states that experiments were "repeated twice," which may not be sufficient for reproducibility in multi-omics comparisons or should be commented.

4. To overcome some of the limitations that might prove to be difficult experimentally, I suggest to include a final paragraph on study limitations that would strengthen the Discussion. It should explicitly acknowledge:

- the restricted number of human models used
- potential confounding due to cancer-derived chromosomal instability,
- the need for validation in primary hepatocytes or organoid systems, and
- that the mechanistic interactions among KDM6A, HNF4A, and CREBH remain partly correlative.

Reviewer #3

(Remarks to the Author)

I would like to thank the authors for their efforts to improve the manuscript. Although they have included new and relevant data, I still have several points that need to be addressed.

In Figure 1J, the authors claim to show that the composition of VLDL is altered. However, Figure 1J presents plasma cholesterol levels during the VLDL secretion assay. Since the cholesterol levels at time point 0 are already much higher in the LKO mice compared to the WT, this conclusion cannot be made. It appears that the relative increase in plasma cholesterol during the poloxamer experiment is similar between the two groups.

To confirm that the VLDL composition is indeed altered, the authors should isolate the VLDL particles by centrifugation and

determine the levels of apoB100, triglycerides (TG), total cholesterol (TC), and phospholipids in these particles. Only then can they conclusively state that the VLDL composition is changed.

In addition, what are the plasma ApoB100 levels in these mice? I expect an increase, which suggests more VLDL particles rather than a change in particle composition.

I disagree that LCAT reduction explains the reduction in plasma TG levels. LCAT deficiency increases plasma TG levels but attenuates diet-induced ApoB-containing lipoproteins.

What is the evidence that bile acid synthesis is reduced, not only in gene expression data? It suggests performing flux analysis of bile acids and determining bile acid concentration over time.

Reviewer #4

(Remarks to the Author)

Version 2:

Reviewer comments:

Reviewer #2

(Remarks to the Author)

In the 2nd revision the authors have successfully addressed most of my remarks. It is important that they acknowledge also the limitations of the study, and by this avoid potential generalisation.

The notion that cholesterol networks differ between males and females is important since this results in marked sex differences in lipoprotein patterns and risk of developing atherosclerosis. The data confirm that KDM6A is a female-biased, X-linked epigenetic regulator essential for hepatic cholesterol and lipoprotein homeostasis, whose loss induces atherogenic lipid profiles specifically in females. The manuscript provides a mechanistic basis, consistent with female protection before menopause and increased vulnerability when protective regulation is lost (even if this is not specifically underlined in the manuscript).

I suggest the paper to be accepted in nature Communications.

Reviewer #3

(Remarks to the Author)

The authors answered my comments sufficiently. I have no further comments.

Reviewer #4

(Remarks to the Author)

General response to the reviewers' comments

We thank all the reviewers for their very supportive comments and appreciate the intriguing issues raised and detailed suggestions made. We have carefully addressed each of these in our extensive experimental revision as outlined in the point-by-point response. We believe that with the inclusion of the revised experimental data we could further improve the quality, the clarity and the conceptual impact of our study.

We would like to highlight the following key improvements of issues raised by all reviewers:

1) We have screened the transcriptomic datasets and identified a suitable male human liver cell line Huh1 for comparable analysis with HRO cells. We have also added karyotype results to our cell lines. We have repeated all the knockdown experiments in the Huh1 cells and found that KDM6A controlled very different transcriptomic signatures between the two cells, with lipoprotein and cholesterol metabolic pathways annotated only in HRO cells but not Huh1 cells, which was consistent with the female specific phenotypes in the LKO mice.

2) We have also performed additional ChIPseq analysis in the Huh1 cells upon knockdown of KDM6A in Huh1 cells and discovered that the overall regulation of KDM6A in the Huh1 cells was also independent of its demethylase activities, despite the major difference in KDM6A gene targets between HRO and Huh1 cells.

3) We have confirmed the coregulation of KDM6A and HNF4A in the Huh1 cells. We found that both transcriptomic and epigenetic changes caused by HNF4A loss were significantly correlated with KDM6A knockdown in Huh1 cells. However, genome widely KDM6A did not require HNF4A to bind to its target genes in Huh1 cells, in contrast to the drastic global KDM6A release upon HNF4A knockdown in the HRO cells.

4) We have performed the transcription factor enrichment analysis in the Huh1 cells and found major difference from HRO cells. CREBH was only enriched in KDM6A and HNF4A coregulated genes in HRO cells but not Huh1 cells. We have in particular confirmed the consistent regulation of KDM6A on CREBH in the female mice by Crebh ChIPseq in both WT and LKO female liver samples. The results showed released binding of Crebh at multiple key cholesterol and lipoprotein metabolic genes such as *Cyp7a1*, *Cyp8b1* and *Lcat*, etc. To our knowledge, this is the first Crebh ChIPseq in mouse liver samples. We believe this dataset will be valuable for further investigation of CREBH metabolic mechanisms in the liver.

5) We have also performed extensive physiological analysis in the mice. We have applied FPLC to analyze the VLDL triglyceride and cholesterol production in the WT and LKO mice. We have also measured the bile acids in the plasma, liver and feces in WT and LKO mice. In combination with the newly added RNAseq results in the AAV-PCSK9 WT and LKO mice, we have proposed mechanisms to explain how disturbed bile acid synthesis, excretion and reuptake affected the cholesterol accumulation in the liver, leading to the enhanced VLDL cholesterol production. We have also identified reduced lipogenesis in the LKO mice liver, which potentially contributed to the slightly reduced lipoprotein and significantly lower liver triglyceride.

Overall, we believe that our work now provides better evidence supporting the sex specificity of KDM6A/HNF4A/CREBH axis in controlling the epigenetic regulation linked with lipoprotein and cholesterol pathways and development of atherosclerosis.

Itemized list of all changes made in the revised manuscript

- 1) Revised **Fig. 1c, 1d, 1o, 1p; Supplementary Fig. 1d, 1e**. We have now included the Huh1 RNAseq analysis to the original HRO RNAseq results.
- 2) Added **Fig. 1i, 1j, 1m, 1n; Supplementary Fig. 2k, 2l, 2m and 2n**. We have performed new RNAseq analysis and analyzed the physiological changes in the AAV-PCSK9 WT and LKO mice, including FPLC measurement of VLDL triglyceride and cholesterol production assay upon lipase inhibition, and bile acid levels in the plasma, liver and feces. We have also validated some key genes using western blot in the mice.
- 3) Added **Fig. 2e, 2f, 2g, 2h, 2m, 2n, 2o, 2p, 2r; Supplementary Fig. 3e, 3f, 3g, 3h, 3q, 3r**. We have performed H3K4me1/2, H3K27ac and H3K27me3 ChIPseq analysis in siKDM6A transfected Huh1 cells. These data showed that KDM6A regulation of its target genes in Huh1 cells did not rely on its demethylase activities, despite that KDM6A targeted genes showed major difference between Huh1 and HRO cells.
- 4) Added **Fig. 3a; Supplementary Fig. 4a, 4b, 4g, 4h, 4i, 4j, 4l, 4m, 4n and 4o**. We have studied the interplay of KDM6A with HNF4A in the Huh1 cells. We found that KDM6A and HNF4A regulated gene expression showed significant correlation, but global KDM6A did not require HNF4A to bind to its target genes in Huh1 cells, unlike in HRO cells. We also found that HNF4A and KDM6A coregulated genes showed major difference between HRO and Huh1 cells, with no pathways annotated to lipoprotein and cholesterol metabolism, unlike in HRO cells.
- 5) Added **Supplementary Fig. 5a, 5b, 5c, 5d, 5e, 5f, 5g, 5h, 5i, 5j, 5k, 5l, 5m, 5n**. We have compared the epigenetic regulation between siKDM6A and siHNF4A in the Huh1 cells and found that HNF4A loss at HNF4A-dependent KDM6A genes reduced H3K4me1/2 and H3K27ac, supporting the non-enzymatic regulation of KDM6A at those loci. However, H3K27me3 was also drastically reduced by HNF4A at those loci, indicating the broader regulation of HNF4A on other epigenetic enzymes such as methyltransferases.
- 6) Added **Fig. 5g, 5h, 5i, 5j; Supplementary Fig. 6a, 6f, 6g, 6h, 6i, 6j, 6k, 6l, 6m, 6n**. We have performed transcription factor enrichment analysis in the Huh1 cells, which showed totally different annotated transcription factors from HRO cells. We have also successfully performed Crebh ChIPseq in the female AAV-PCSK9 WT and LKO liver samples for the first time and confirmed the species conserved mechanism of attenuated Crebh binding at key lipoprotein and cholesterol metabolic genes upon *Kdm6a* loss.

The original sequencing data are uploaded to GEO database with the link and token as follows:

RNA-seq Data: <https://www.ncbi.nlm.nih.gov/geo/query/acc.cgi?acc=GSE287688>
Secure token of RNA-seq Data(GSE287688) for reviewers: uhglgqesbtsdjsj

CUT&Tag Data: <https://www.ncbi.nlm.nih.gov/geo/query/acc.cgi?acc=GSE287680>
Secure token of CUT&Tag Data(GSE287680) for reviewers: ezyncmgcjfexfeb

ChIP-seq Data: <https://www.ncbi.nlm.nih.gov/geo/query/acc.cgi?acc=GSE287736>
Secure token of ChIP-seq Data(GSE287736) for reviewers: qpkpaceolxithix

To be noted, we have been informed that the update of the files in the GEO system might be delayed due to ongoing maintenance.

Point-by-point response to the reviewer's comments

Our responses are indicated in RED.

REVIEWER COMMENTS

Reviewer #2 (Remarks to the Author):

The manuscript studies an underrepresented area – the molecular basis beyond the sexual dimorphism of the liver. The rationale of the study is a known phenomenon that the expression of KDM6A (Lysine Demethylase 6A) differs between females and males and that its expression can be lost in liver cancers. KDM6A is located on X chromosome and since it can escape X inactivation it is expressed higher in female compared to the male tissues. By integrating transcriptomic and epigenomic techniques in human liver cell lines and liver-specific knockout mice and disease models, the authors aimed to investigate the physiological roles and regulatory mechanisms of the hepatic KDM6A. They promise to answer the question what is the metabolic function of KDM6A in regulating liver lipid metabolism, how such regulation is modulated in the chromatin epigenetic levels and whether liver KDM6A deficiency leads to sex difference in developing metabolic disorders. The questions are relevant and novel but I am not sure that the answers provided are always clear and sufficiently general. I question also whether the sex dimorphism has been proven for the KDM6A-HNF4A-CREBH network.

To draw the conclusions the authors used two hepatic cell lines of different sexes (female line HROHep03 and male cell line Huh7, both from Cytion) HROHep03 is a primary liver adenocarcinoma cell line; Huh7 is a well-differentiated hepatocellular carcinoma (HCC) cell line. They also used mouse strains, Kdm6flox/flox developed by Cyagen and WT controls, with or without the mPCSK9 delivered by adenovirus.

Major remarks.

Studies in cell lines: While both HROHep03 and Huh7 cell lines originate from liver cancers in individuals of different sexes, their cancerous nature and associated chromosomal abnormalities complicate the assessments of X-chromosome gene dosage. For precise determination, experimental analyses such as karyotyping, quantitative PCR, or RNA sequencing would be necessary to assess the X-chromosome content and gene expression levels in these cell lines. This would prove whether these two cell lines are good models to study KDM6A in relation to sex dimorphic differences in metabolic pathways, potentially leading to disorders. The authors should thus determine the X-chromosome and KDM6A gene dosage of both cell lines to illuminate some of their conclusions.

Reply: We appreciate the suggestions by reviewer 2. We have analyzed the X and Y chromosome content according to the reviewer's suggestion by real-time PCR in cell lines which were used in the revised manuscript. We used early passage primary human aortic smooth muscle cell (HASMC) as a control (**Supplementary Fig. 1c**). Our result showed that HRO-Hep03 cells have 2 copies of X chromosomes, Huh7 cells and Huh1 cells (a new cell line used in the revised manuscript) have 1 copy of X chromosome (**Supplementary Fig. 1c**). The result is now added in the revised manuscript.

However, we have unexpectedly found that the Y chromosome is absent in the Huh7 cells (**revised Supplementary Fig. 1b**). This finding was supported by our own RNAseq data (**Rebuttal Fig. 1b**). Similar finding was reported previously by karyotyping studies from other groups¹, which showed no Y chromosome in Huh7 cells using FISH, despite that the cell line was derived from a male patient. The public available STR analysis from human cell lines (<https://www.cellosaurus.org/>) further confirmed the absence of Y chromosome in the Huh7 cells. This issue will be further discussed in the rebuttal letter.

Rebuttal Figure 1: Genome browser screenshot of RNAseq in male human liver cells at a) *KDM6A/UTX*, b) *KDM6C/UTY*, c) *CREBH* and d) *HNF4A* gene loci. And in male mouse liver samples at e) *Kdm6a/Utx* and f) *Kdm6c/Uty* loci.

For example, the gene ontology analysis in KDM6A knockdown male and female liver cell lines showed similar cholesterol and lipoprotein modulating pathways, which is unexpected taking in consideration the envisioned higher expression of KDM6A in females.

Reply: We appreciate this very important issue raised by reviewer 2. Our RNAseq analysis in siKDM6A knockdown HRO and Huh7 cells indeed showed significant coregulated genes and identical gene ontology annotations. We believe that this is because Huh7 lacks KDM6C/UTY, which is a Y-linked histone demethylase belonging to the KDM6 family together with KDM6A/UTX. The two enzymes share similar domains and structures, and partially compensate each other's functions. Both male human and mice liver cells express high and identical levels of KDM6A and KDM6C (**Rebuttal Fig. 1a,1b,1e,1f**). Due to lack of Y chromosome and its linked KDM6C/UTY in the Huh7 cells, the compensatory effects of KDM6C/UTY to KDM6A loss is missing. This explains why knocking down of KDM6A in the Huh7 cells showed similar transcriptional signatures with HRO-Hep03 cells, annotated to lipoprotein and cholesterol metabolic pathways.

In the revised manuscript, we reviewed the public RNAseq data from different liver cell lines (GSE97098) and 4 separate male primary human hepatocytes (our own data, **Supplementary Fig. 1a**). Using *KDM6A* and *KDM6C* mRNA expression levels and the *KDM6C/KDM6A* ratio as two major parameters, we have identified Huh1 and SNU182 cell lines as potential models for our

study (**revised Supplementary Fig. 1a**). Both cell lines have identical *KDM6A* and *KDM6C* expression levels and ratio comparing to male primary human hepatocytes (**Supplementary Fig. 1a**). We have further checked the mRNA expression of *HNF4A* and *CREBH*, and found SNU182 cell line did not express *HNF4A* or *CREBH* (**Rebuttal Fig. 1c, 1d**). we therefore repeated all the cell experiments using Huh1 cell line instead in our revised manuscript (discussed later in the rebuttal letter).

For some later mechanistic studies it is not always clear why sometimes only HRO cell have been used while in other cases only Huh7. If any comparison is to be made one would expect parallel experiments in both cell lines, even if they are not optimal. Or at least a more structured representation of experiment and data flow, considering the sex dimorphism in the focus.

Reply: Thanks for the suggestions from reviewer 2. In the revised manuscript, we have performed substantial experiments in Huh1 cells and arranged the data alongside the HRO results, these experiments are listed as follows:

1) We have performed RNAseq in siKDM6A transfected Huh1 cells to compare the transcriptomic targets of KDM6A between the cells (**Fig.1b,c,d; Supplementary Fig. 1d, e**).

2) We have performed H3K4me1, H3K4me2, H3K27ac and H3K27me3 ChIPseq analysis in siKDM6A transfected Huh1 cells to study the epigenetic signatures linked with KDM6A loss in the Huh1 cells (**Fig. 2e,f,g,h,m,n,o,p,r, Supplementary Fig. 3e,f,g h,q,r**).

3) We have done RNAseq in siHNF4A transfected Huh1 cells and compared the transcriptomic signatures with siKDM6A treated cells (**Supplementary Fig. 4k, l, m**).

4) We have performed KDM6A CUT&Tag in siHNF4A knockdown Huh1 cells to study the chromatin interplay of HNF4A and KDM6A in the Huh1 cells (**Supplementary Fig. 4a,b, g, h, i, j, p**).

5) We have also compared the epigenetic changes by HNF4A and KDM6A in the Huh1 cells by ChIPseq of H3K4me1, H3K4me2, H3K27ac and H3K27me3 in siHNF4A transfected Huh1 cells (**Supplement Fig. 5a,b,c,d,e,f,g,h,I,j,k,l,m,n,o**).

We have now arranged all the data from HRO and Huh1 cells in parallel in the revised manuscript.

Studies in the mouse models: Authors showed that the atherogenic lipoprotein profiles were only observed in female but not male KDM6A LKO mice. Was the background of the KDM6A flox/flox mice 100% B6? Since for B6 the escape from X-inactivation has been documented, but I am not sure for other mouse lines.

Reply: Our floxed mice were bred with C57BJ6 mice for 9 generations before crossing with the *Albumin-Cre* mice. The *Albumin-Cre* strain was originally from Jackson lab but has been bred with C57BJ6 mice for 9 generations before embryo freezing and cross breeding with the *Kdm6a* floxed mice in this study. We have also performed real-time PCR to verify the expression of *Kdm6a* in our male and female mice (**Rebuttal Fig. 2**). This information is now added to the methodology part.

Rebuttal Figure 2: Real time PCR of *Kdm6a* mRNA in male and female mouse liver samples (n=4 in each group). Student's t test $* < 0.05$

Chip-Seq – why was the Chip-seq analysis for methylation and acetylation KDM6A activities performed only in the HRO cells? Is it true also for Huh7 cells (or male cells) that KDM6A regulation of lipoprotein metabolic genes is independent of H3K27me3 demethylation? I am not convinced that the Chip-seq data in the female floxed and LKO mouse liver data are confirmatory of the HRO cell data.

Reply: Thanks for the suggestions. In the revised manuscript, we have repeated all the RNAseq and ChIPseq experiments in Huh1 cells. Knocking down of KDM6A in Huh1 cells had very few coregulated genes with HRO cells (**Supplementary Fig. 1e**). Gene ontology in Huh1 cells also showed no lipoprotein and cholesterol metabolic pathways, which was very different from HRO cells (**Fig. 1c, Supplementary Fig. 1d**). This is further supported by the genome browser screenshot at *APOA1/APOC3/APOA4* cluster (**Fig. 2r**), *APOM* (**Supplementary Fig. 3q**) and *APOH* (**Supplementary Fig. 3r**) loci, in contrast to the substantial changes at the same loci in the HRO cells (**Fig. 2q, Supplementary Fig. 3o, 3p**).

In order to further investigate whether the transcriptomic changes in both HRO and Huh1 cells depend on H3K27me3. We have instead compared the correlation of all the differentially expressed genes ($p_{adj} < 0.05$) in HRO and Huh1 cells with the epigenetic markers without limiting the gene signature to the lipoprotein or metabolic pathways (**Fig. 2i-2p**). The result consistently showed that in both HRO cells and Huh1 cells, the transcriptomic changes upon KDM6A loss were significantly correlated with H3K4me1,2 and H3K27ac but not H3K27me3 (**Fig. 2i-2p**). Indeed, at Huh1-specific KDM6A target genes, we consistently observed reduced H3K4me1,2 and H3K27ac changes (**Rebuttal Fig. 3a, 3b**). These results supported that in the genome-wide level, the KDM6A mediated transcriptomic regulation does not rely on H3K27me3.

Rebuttal Figure 3: Genome browser screenshot of H3K4me1, 2, H3K27ac and H3K27me3 in control and KDM6A knockdown Huh1 cells at a) *SLC6A19* and b) *TRABD2B* loci.

We agree with the reviewer that the ChIPseq data from the female LKO mice was not entirely confirmatory to the HRO cells. This might partially be due to the difference between species as well as between primary cells and liver cell lines as pointed out by reviewer 2 in the first comment, which we unfortunately cannot control in our system.

For example, knocking down of KDM6A in the HRO cells did not trigger global changes in H3K4me1,2 or H3K27me3 but only affected H3K27ac minorly (**Fig. 2a-2d, Supplementary Fig. 3a-3d**). While in LKO female mice, H3K4me1 was already reduced both genome-widely and at KDM6A target gene loci (**Supplementary Fig. 3s-3v**). However, this LKO data supported that the transcriptomic regulation in LKO might be less related with H3K27me3 changes and therefore did not require H3K27me3 demethylase activity of KDM6A, consistent with the findings in HRO cells. We also looked at *Apoa1/Apoc3/Apoa4* cluster and *Apom* loci in the LKO liver, which showed similar effects in H3K4me1,2, H3K27ac but not H3K27me3 (**Supplementary Fig. 3u-3v**). These results also confirmed the findings in HRO.

Moreover, in the revised manuscript, we analyzed the transcription factor enrichment in the KDM6A target genes in LKO mice. Like HRO cells, CREBH appeared also on the top list (**Fig. 5g**). We further performed CREBH ChIPseq in LKO mice. Our data showed that CREBH binding was reduced both globally and at KDM6A target gene loci in the LKO liver (**Fig. 5i-5j**), again consistent with findings in the HRO cells (**Fig. 5e-5f**).

Collectively, based on our current data, despite that the KDM6A regulated epigenetic changes differed in range and gene loci between mouse liver and human liver cell lines, the key findings that such regulation relied more on the non-enzymatic activities of KDM6A and that the remodeled chromatin microenvironment led to differential chromatin affinity of CREBH transcription factor to lipoprotein and cholesterol metabolic genes, are consistent between the LKO and HRO cells.

To identify the TFs responsible for KDM6A recruitment to the lipoprotein and cholesterol (LnC) genes - HNF4A knockdown reduced KDM6A peaks in both Huh7 and HRO cells, so based on the cell lines, this is a general and not sex-specific event.

Reply: Thanks for the insightful comments. We indeed observed loss of KDM6A peaks in siHNF4A in both HRO and Huh7 cells. As we described above, Huh7 turned out to lack Y chromosome. It is therefore not surprising that Huh7 behaved more like HRO in KDM6A regulated genes and pathways, as both cell lines lack Y-linked KDM6C to compensate the loss of KDM6A in both cell lines.

In the revised manuscript, we have repeated KDM6A CUT&Tag in siHNF4A transfected Huh1 cells. The intensity plot showed that loss of HNF4A had no effects on global KDM6A binding in Huh1 cells (**Supplementary Fig. 4g-4h**), which was very different from HRO or Huh7 cells (**Fig. 3c-3d**). We further checked the HNF4A and KDM6A coregulated genes in Huh1 cells. Despite that almost 25% of the KDM6A target genes are coregulated by HNF4A (**Supplementary Fig. 4k**) in the Huh1 cells, the KDM6A/HNF4A common genes in the Huh1 cells are very different from the HRO cells, with no lipoprotein or cholesterol metabolic pathways annotated in the gene ontology analysis (**Supplementary Fig. 4l-4m**). We also found very little changes in KDM6A recruitment at HNF4A/KDM6A coregulated gene loci in siHNF4A Huh1 cells (**Supplementary Fig. 4o**) in contrast to the significant loss of KDM6A peaks upon HNF4A knockdown in HRO cells (**Supplementary Fig. 4n**).

Rebuttal Figure 4: Hypothetic model that KDM6A and KDM6C are recruited by HNF4A to control *APOA1* expression.

However, at gene loci such as *Apoa1/Apoc3/Apoa4* gene cluster, knockdown of HNF4A reduced KDM6A binding (-75% at *Apoc3* promoter, -40% in *Apoa1* promoter). While knockdown of KDM6A alone did not change H3K4 or H3K27 epigenetics, nor the mRNA expression of *Apoc3* or *Apoa1* in Huh1 cells (**Supplementary Fig. 4j**). We hypothesize that both KDM6A and KDM6C colocalize with HNF4A at some gene loci. While removal of KDM6A is compensated by KDM6C at the same locus, knockdown of HNF4A releases both KDM6A and KDM6C (**Rebuttal Fig. 4**). This model was supported by our data that knockdown of HNF4A but not KDM6A in the Huh1 cells led to significant reduction of H3K4me1,2 and H3K27ac at *Apoa1* loci in the Huh1 cells (**Supplementary Fig. 5m**).

Collectively, our data showed that the interplay of HNF4A and KDM6A was sex-specific (different in male Huh1 and female HRO cells) in the genome wide level. Despite that KDM6A requires HNF4A to bind to some gene loci such as *Apoa1* in both HRO and Huh1 cells, removal of KDM6A did not necessarily lead to epigenetic and transcriptomic changes in the Huh1 cells unlike in HRO cells due to the compensatory effects of KDM6C at the same loci.

Suppl. Fig 4 (j) shows the heatmap that compares the downregulated genes in KDM6A LKO female mice and HNF4A LKO male mice (GSE173968). This is not specified in the text of the main paper: "We also compared liver RNA-seq of the female LKO mice with HNF4A LKO mice (GSE173968)39, and found that around 30% of the downregulated genes in the KDM6A LKO mice were overlapping with HNF4A LKO". Claiming that the data was consistent with the HRO results does not prove that this is a sex dimorphic event.

Reply: Thanks for raising this issue. Despite that knockdown of KDM6A had very few overlapped downregulated genes between HRO and Huh1 cells (**Supplementary Fig. 1e**), knockdown of HNF4A in both cell lines showed a much higher percentage of overlap (**Rebuttal Fig. 5**). This can be explained by the above model (**Rebuttal Fig. 4**) that removal of HNF4A abrogated both

Rebuttal Figure 5: Venn diagram of downregulated genes by siHNF4A in HRO and Huh1 cells.

KDM6A and KDM6C and therefore had less sex-specific transcriptomic changes comparing to KDM6A loss. Therefore, we thought it was meaningful to compare the KDM6A target genes in the female LKO with male HNF4A LKO transcriptome as there is currently no HNF4A LKO female transcriptome available in the public database.

However, we understand the concern of reviewer 2 because so far we cannot directly prove the above model due to lack of validated tools (i.e. ChIPseq quality KDM6C antibodies). We therefore decide to remove this supplementary figure in the revised manuscript.

KDM6A removal affects the binding and transcription activation of CREBH- the RNA-seq data from HRO and Huh7 cells identified several common TFs, where among the top candidates was also CREBH. Again, where is the sex dimorphism?

Reply: We have repeated our experiment in Huh1 cells. Knockdown of KDM6A caused very different transcriptomic changes in the Huh1 cells comparing with HRO cells. The same transcription factor analysis in the Huh1 cells also showed a totally different transcription factor list, with a group of ZNF factors on the top list (**Supplementary Fig. 6a**).

The title is over-ambitious since sex dimorphism was not experimentally proven for all components promised in the title.

Reply: Thanks for raising the concerns. We understand the word ‘dimorphism’ is over generalized and does not precisely reflect the experimental evidence. Our revised manuscript has provided evidence that:

1) *KDM6A* deletion affected lipoprotein and cholesterol metabolic genes only in the female liver cell line and female LKO mice but not in the males.

2) Liver *Kdm6a* knockout affected lipoprotein cholesterol levels only in the female but not in the male mice.

3) HNF4A deletion changes global KDM6A binding only in the female but not male liver cell line. Even at certain gene loci (*APOA1/APOC3*), HNF4A was required for KDM6A binding for both male and female liver cells, KDM6A loss only affected epigenetics and gene expression in the female but not male liver cells possibly due to compensation from KDM6C.

4) The interplay of KDM6A and HNF4A regulated CREBH binding only in the female but not male liver cells.

Based on these data, we believe that the new title ‘**Sex-specific KDM6A-HNF4A-CREBH network controls lipoprotein and cholesterol metabolism and atherosclerosis via epigenetic reprogramming of hepatocytes**’ more precisely summarizes the findings of this revised manuscript.

The statement “We have therefore identified a novel sex-specific cholesterol regulatory mechanism in the liver.” is over ambitious. The authors have to specify clearly what is indeed sexually dimorphic (and what common to both sexes), and based on which model the conclusions have been made. It is not a problem that the data from cell lines do not always correlate with the mouse data. The problem is “generalization” of findings in cases when the experiments do not support the case.

Reply: Thanks for this very fair comment. We have changed the statement in the introduction to ‘We have therefore identified a cholesterol regulatory mechanism controlled by X-linked epigenetic enzyme KDM6A in the liver’, which is more descriptive and less general. We have further included in the discussion part more detailed description of our findings.

Minor comments:

- The background line of the *Kdm6a*^{flox/flox} mice is not described.

Reply: We have included the information in the methodology part of the revised manuscript.

-There is no data availability statement and location where transcriptome/ChIP-seq is deposited.

Reply: We have included the GSE numbers in the revised manuscript.

-Some smaller text corrections/typos on page 6 line 4,7, page 8 line 14, page 10 line 22 and page 12 line 14.

Reply: The errors are now corrected in the revised manuscript.

-In the Method part CUT&Tag sample preparation authors state "DNA was purified using SPRI beads and quantified using a fluorometric method". Detail on "fluorometric method" should be added. It should also be stated what is the origin of "50.000 harvested cells".

Reply: The fluorometric method refers to the Qubic fluorometer (Thermo Fisher). The measurement was done following the protocol of the equipment.

The 500.000 cells are HRO or Huh1 cells.

All the above information is now added to the revised manuscript.

Reviewer #3 (Remarks to the Author):

In this study, the authors demonstrated that KDM6A regulates the expression of lipoprotein metabolism-related genes in a sex-specific manner. Using mice, they showed that hepatic loss of KDM6A results in elevated plasma cholesterol levels and accelerates the development of atherosclerosis in female mice but not in male mice. KDM6A is recruited to chromatin by HNF4A and, through CREBH, activates the transcription of lipid-related genes.

This is a very interesting study, clearly presented and well-written. However, I have some comments that could help improve this study.

In Figure 1, the authors showed that the loss of KDM6A leads to a decrease in mRNA levels encoding proteins involved in lipid and cholesterol metabolism, such as LIPC, HMGCR, and MTTP. This downregulation was conserved between human cells and mouse livers.

Interestingly, despite the downregulation of many lipid-related genes, plasma lipid levels increased. In the absence of LDLR, there was a two-fold increase in VLDL. How is this possible? As shown in Supplemental Figure 1i, genes involved in VLDL biogenesis were decreased. Do the authors have an explanation for this discrepancy?

Reply: We appreciate this insightful comments from the reviewer 3. In fact, the plasma VLDL triglyceride level was slightly (although not significantly) decreased in the female LKO mice (**Fig. 1g**), but the VLDL cholesterol level increased more than 2 folds (**Fig. 1e**). In consistence, the intrahepatic triglyceride level was also significantly lower in female LKO mice (**Fig. 1l**), in contrast to the intrahepatic total and free cholesterol in female LKO mice (**Fig. 1k**). These data suggested changed VLDL composition in the LKO mice, with higher cholesterol but lower triglyceride levels.

In order to identify the mechanisms underlying this phenotypical change, we performed an additional RNA-seq in WT and LKO mice after PCSK9 injection and western diet treatment. The GO analysis in the LKO downregulated genes showed enrichment in cholesterol metabolic pathways (**Fig. 1n**), consistent with the in vitro data in the human HRO liver cell lines (**Fig. 1c**). In particular, we have observed significant downregulation of *Cyp7a1*, *Cyp8b1* and *Akr1d1*, which are key enzymes regulating bile acid synthesis. Downregulation of these genes decreases the cholesterol clearance in the liver and promotes liver cholesterol accumulation, which we have seen in the female LKO mice (**Fig. 1k**). *Lcat* was also significantly downregulated in the LKO mice. Because *Lcat* controls free cholesterol esterification, downregulation of *Lcat* may further promote liver free cholesterol accumulation and affect reverse cholesterol transport from HDL back into

the liver, leading to increased VLDL/LDL cholesterol as we have seen in the LKO female mice (**Fig. 1e-1f**). Downregulation of these above genes in combination changed liver and plasma cholesterol levels and either directly or indirectly affected VLDL cholesterol composition. There are also genes involved in bile acid excretion and reuptake in the LKO female mice. For example, *Slco1b2* encodes Oatp1b2 which is a sodium independent transporter involved in uptake of bile acids from the blood. It was also downregulated in the LKO female mice, as a result, we have observed more than twice increase in blood bile acid levels in the LKO female mice (**Supplementary Fig. 2k**). We have also found bile acid processing genes such as *Ugt1a9*, *Abcc6*, *Sult2a3*, *Sult2a5*, *Sult2a8*, *Cyp2c37*, etc. Downregulation of those genes in theory affects excretion of bile acids, leading to accumulated bile acids in the liver. As a result, despite reduced bile acid synthesis rate due to downregulated *Cyp7a1*, *Cyp8b1* and *Akr1d1*, the bile acids still accumulated in the LKO female mice (**Supplementary Fig. 2k, 2l**).

For the triglyceride levels, we have seen *Srebp1* and *Dgat2* expression significantly downregulated in the liver, suggesting reduced lipogenesis. Indeed, the liver triglyceride level was significantly lower in LKO female mice (**Fig. 1i**) and the VLDL triglyceride was even slightly lower in the LKO female mice as well.

Based on these data, we believe that change VLDL triglyceride and cholesterol was due to the altered lipid and cholesterol metabolism in the liver which thereby affected their composition in the VLDL production. This theory was further supported by the VLDL production assay in which we applied FPLC to all the plasma fractions from the PCSK9 and diet challenged WT and LKO mice at different time points upon inhibiting the lipase activity (**Fig. 1i and 1j**). The results precisely showed no significant change in VLDL triglyceride output but higher cholesterol levels, proving that the changes in VLDL was due to altered liver lipid and cholesterol output.

Moreover, our revised manuscript has also confirmed the consistent contribution of *Crebh* in regulating the above genes (**Fig. 5g-5j**, **Supplementary Fig. 6f-6n**). To be noted, liver KO of *Crebh* in mice showed increased VLDL and decreased HDL cholesterol, which was partially consistent with the phenotypes in our *Kdm6a* LKO mice.

We have now summarized the transcriptomic changes and potential physiological impacts in (**Rebuttal Fig. 6**). We believe these genes contributed to the altered liver and plasma triglyceride and cholesterol levels in the LKO mice in combination.

The study mainly focuses on gene expression, but did the authors also observe effects at the protein level? Is the increase in VLDL due to enhanced VLDL biogenesis, increased cholesterol synthesis, or a defect in clearance? Essential pathways that has to be studied to explain the mechanism underlying the lipid phenotype, as the lipid phenotype cannot be fully explained by the gene expression data.

Reply: Thanks for the suggestions from the reviewer. We have performed comparative proteomics to confirm the results we found in the RNAseq in the protein level. We have indeed observed the same trend of changes in some key genes in the proteomics results (**Rebuttal Fig. 7**). However, because the resolution of the proteomics is much lower than RNAseq, we were not able to catch all the genes through proteomics. Also due to the high variation between each sample in the proteomics analysis, we were only able to get statistical significance in very few genes in total, we therefore used western blot to confirm the expression difference in several but not all the key proteins. We were able to see clear reduction of Cyp7a1, Srebp1 between WT and LKO female mice (**Supplementary Fig. 2n**). We have included the western blot into the revised manuscript.

We have explained the mechanisms how VLDL composition was changed in the LKO female mice. We have also performed VLDL production assay in the revised manuscript to prove the theory that the altered liver lipid and cholesterol metabolism contributed to the VLDL production changes in the LKO mice (**Rebuttal Fig. 6**).

From the RNAseq level, we were able to explain that the differential levels of VLDL triglyceride and cholesterol levels in the LKO female mice was due to changed liver cholesterol and triglyceride levels, caused by impaired cholesterol clearance (reduced bile acid synthesis and excretion) and decreased lipogenesis in the liver (See reply to the above comment).

Rebuttal Figure 7: a) Proteomics and b) western blot analysis of the PCSK9+WD treated WT and LKO female liver samples.

Similarly, for liver lipid levels, there is no clear correlation between the decrease in lipid-related genes and liver lipid accumulation. Is lipogenesis increased, or beta-oxidation?

Reply: We appreciate this important point raised by the reviewer 3. From the RNAseq results between the PCSK9 and western diet treated LKO and WT groups, the mRNA levels of both *Srebf* and its chaperone *Scap* and target gene *Dgat2* were downregulated in the LKO mice, suggesting that the reduced liver triglyceride level was likely contributed by decreased lipogenesis (**Rebuttal Fig. 6**). We have now included this into the results and discussion part of the revised manuscript.

Next, the authors showed that HNF4A is required for the recruitment of KDM6A to lipid gene loci. In Supplemental Figure 4j, they demonstrated an overlap in genes regulated by KDM6A and HNF4A in the mouse liver. What is the effect of HNF4A downregulation on plasma and liver lipids? Do these two models exhibit the same phenotype?

Reply: Thanks for the suggestions from the reviewers.

We have compared the phenotypes of HNF4A LKO²⁻⁴ and KDM6A LKO female mice, there are indeed some similar phenotypes. For example, both strains showed increased liver cholesterol accumulation (partially due to decreased *Cyp7a1* expression and bile acid synthesis)³, and reduced HDL cholesterol (due to decreased expression of *ApoA1/2*)⁴. The VLDL profile changes between *Hnf4a* LKO and *Kdm6a* LKO mice are different because *Hnf4a* regulates a group of VLDL assembling genes such as *Apob100*, and knockout of *Hnf4a* leads to a much stronger reduction of *Mttp* (similar to *Kdm6a* LKO which also shows a mild decrease)², therefore *Hnf4a* LKO sequesters the VLDL in the liver, leading to accumulation of intrahepatic triglyceride and cholesterol but reduced blood VLDL triglyceride and cholesterol levels⁴. Apart from that, *Hnf4a* but not *Kdm6a* controls expression of *Ldlr*⁴, which is a major regulator of blood LDL cholesterol. Overall, *Hnf4a* LKO has a much stronger phenotype comparing to *Kdm6a* LKO.

From our RNA-seq comparison results in HRO cells (**Supplementary Fig. 4k**), we have observed a much broader regulatory targets of HNF4A comparing to KDM6A. This is also supported by the ChIP-seq results that the binding of HNF4A is almost 4 times of KDM6A in the human liver cells (**Supplementary Fig. 4d**), suggesting a much broader regulatory function of HNF4A in the liver. While KDM6A requires HNF4A to bind to key cholesterol metabolic genes, HNF4A has been also reported to interplay many more important lipid and cholesterol transcriptional modulators such as GR, LXR, PPAR, etc, which explains the much broader effects of *Hnf4a* knockout in the liver.

We have included this information in the discussion of the revised manuscript.

Additionally, liver-specific CREBH-deficient mice share a similar phenotype with KDM6A LKO mice. The authors discuss the HDL phenotype but not the VLDL/LDL phenotype. Can they speculate on the underlying mechanism? The gene expression data do not seem to fully support the proposed explanation.

Reply: Thanks for the comments from reviewer 3. There is some similarity and difference in VLDL/LDL phenotype between *Crebh* and *Kdm6a* KO mice. Both mice showed increased VLDL cholesterol levels in the *Ldlr* deletion background. And knockout of both *Crebh* and *Kdm6a* promoted the progression of atherosclerosis in the *Ldlr* deleted mice. While in *Crebh* KO mice, the VLDL triglyceride was also increased while the *Kdm6a* LKO mice showed slight reduction in VLDL triglyceride⁵. Moreover, liver-specific overexpression of *Crebh* in the mice reduced both VLDL triglyceride and cholesterol, further confirming the phenotype⁶. Several mechanisms were proposed to explain the *Crebh* KO or overexpression phenotypes. It was proposed that *Apoa4* was the main regulator involved in VLDL regulation of *Crebh* KO mice. As *Apoa4* coactivates lipid lipase, its downregulation reduced lipid lipase activities, leading to accumulated total VLDL particles in the blood⁵. While in *Kdm6a* LKO mice, we have performed VLDL production assay by inhibiting the lipase activities in the mice. Our result showed unchanged production of VLDL triglyceride but significant increase of VLDL cholesterol in the mice (**Fig. 1e-1h**), suggesting that the lipoprotein changes in *Kdm6a* LKO mice was not related with the VLDL clearance or altered lipase activities but rather differential production of triglyceride and cholesterol from the liver. It was also proposed that the triglyceride and cholesterol lowering effects of *Crebh* was through *ApoE*, as *Crebh* overexpression in the liver significantly induced *ApoE* expression and the protective role of *Crebh* was gone in *ApoE* KO mice⁶. Neither *Apoa4* nor *ApoE* expression levels were changed in *Kdm6a* LKO mice. These data suggested that although CREBH and KDM6A shared common regulatory genes, their transcriptomic target signatures are not entirely the same and therefore the knockout of both proteins in the liver may possess specific regulatory functions.

However, from our RNAseq results, we did identified multiple common genes involved in lipoprotein metabolism by both Kdm6a and Crebh. One of the commonly regulated genes was *Cyp7a1*, a rate-limiting enzyme involved in bile acid synthesis. *Crebh* knockout also reduced *Cyp7a1* mRNA expression, which potentially induced liver cholesterol accumulation in the knockout mice⁷. To be noted, we were able to prove for the first time by ChIPseq that *Crebh* directly bound to *Cyp7a1* and *Kdm6a* loss attenuated such binding, leading to decreased *Cyp7a1* mRNA levels in the LKO mice (**Supplementary Fig. 6e**).

Collectively, KDM6A and CREBH shared commonly regulated cholesterol metabolic genes which potentially and partially contributed to the shared phenotypes of both knockout mice strains. However, the transcriptomic targets of KDM6A and CREBH are not entirely the same, explaining the differential mechanisms related with VLDL/LDL regulation. We have now included this into the discussion.

Minor points:

Page 8: "...while at adjacent CLPTM1 loci (?):" This appears to be an editing error that was not removed before submission.

Reply: The error is now corrected in the revised manuscript.

Page 10: "KDM6A interplays [interacts?] with..."—this may be an editing error.

Reply: The error is now corrected in the revised manuscript.

Page 12: "KDM6A in regulating (??):" This also seems to be a typo.

Reply: The error is now corrected in the revised manuscript.

Reviewer #4 (Remarks to the Author):

Reply: Many thanks for the input and suggestions.

References:

1. Kasai, F., Hirayama, N., Ozawa, M., Satoh, M. & Kohara, A. HuH-7 reference genome profile: complex karyotype composed of massive loss of heterozygosity. *Hum. Cell* **31**, 261–267 (2018).
2. Hayhurst, G. P., Lee, Y.-H., Lambert, G., Ward, J. M. & Gonzalez, F. J. Hepatocyte Nuclear Factor 4 α (Nuclear Receptor 2A1) Is Essential for Maintenance of Hepatic Gene Expression and Lipid Homeostasis. *Mol. Cell. Biol.* **21**, 1393–1403 (2001).
3. Inoue, Y. *et al.* Regulation of bile acid biosynthesis by hepatocyte nuclear factor 4 α . *J. Lipid Res.* **47**, 215–227 (2006).

4. Yin, L., Ma, H., Ge, X., Edwards, P. A. & Zhang, Y. Hepatic Hepatocyte Nuclear Factor 4 α Is Essential for Maintaining Triglyceride and Cholesterol Homeostasis. *Arterioscler. Thromb. Vasc. Biol.* **31**, 328–336 (2011).
5. Park, J.-G., Xu, X., Cho, S. & Lee, A.-H. Loss of transcription factor CREBH accelerates diet-induced atherosclerosis in *Ldlr*^{-/-} mice. *Arterioscler. Thromb. Vasc. Biol.* **36**, 1772–1781 (2016).
6. Shimizu-Albergine, M. *et al.* CREBH normalizes dyslipidemia and halts atherosclerosis in diabetes by decreasing circulating remnant lipoproteins. *J. Clin. Invest.* **131**, (2021).
7. Chanda, D. *et al.* Hepatic Cannabinoid Receptor Type 1 Mediates Alcohol-Induced Regulation of Bile Acid Enzyme Genes Expression Via CREBH. *PLoS ONE* **8**, e68845 (2013).

General response to the reviewers' comments

We thank all the reviewers for their insightful comments and suggestions. We have carefully addressed the issues raised by the reviewers with more experiments or discussions, as outlined in the point-to-point responses. We believe that the quality and clarity of the revised manuscript has been improved.

We hereby highlight the following key improvements of issues raised by the reviewers.

1) We have further analyzed the VLDL particle number using a state-of-the-art biophysical technique called Single-Particle Profiling (Taras Sych, et al. Nature Biotechnology 2024) via collaboration. Comparing with the traditional APOB100 ELISA (which we also presented in the revised manuscript), the SPP could precisely quantify the VLDL particle numbers. The data supported our previous conclusion that the VLDL particle numbers were not altered in the LKO mice.

2) We have also analyzed the VLDL cholesterol to triglyceride ratio using the precise simultaneous analysis of both cholesterol and triglyceride in the FPLC separated VLDL fraction in the mouse plasma samples. The data clearly showed increased cholesterol to triglyceride ratio in the LKO mice, further supporting the changed composition of VLDL in the LKO mice in the manuscript.

3) We have measured the bile acid composition in the WT and LKO female mice. The data showed more than 10 times increase in conjugated bile acids and marginal difference in unconjugated bile acids in the female LKO mice. These data clearly suggested bile acid excretion defects in the female LKO liver and provided a strong mechanistic explanation to the accumulated liver cholesterol and bile acids which was also consistent with the transcriptomic changes.

4) We have also measured C4 as a surrogate marker for bile acid synthesis in the WT and LKO mouse plasma in the revised manuscript. The result is discussed in the revised manuscript. The reason why C4 and bile acid flux analysis may not be used to accurately measure the bile acid synthesis in the female LKO mice was also explained in the rebuttal letter and revised manuscript.

5) We have performed extra biological replicates for CREBH ChIP-seq in HRO and female LKO mouse liver samples. All the OMICs data in the revised manuscript now contain at least 3 biological replicates. We have edited the methodology part to enhance its clarity.

6) We have formulated the language in some of the terms according to the reviewer's suggestions (i.e. from 'male and female liver cell lines' to 'liver cell lines from male and female patients', etc). We have also included a session 'limitation of the study' to the discussion part to notify our readers the major limitations of our current manuscript.

Overall, we believe that our work now provides better evidence supporting the sex specificity of KDM6A/HNF4A/CREBH axis in controlling the epigenetic regulation linked with lipoprotein and cholesterol pathways and development of atherosclerosis.

Itemized list of added and revised figures and supplementary figures in the manuscript

- 1) Added Fig. 1k, Supplementary Fig. 2i, 2j, 2p, 2q, 2r, 2s, 2t
- 2) Revised Fig. 5c, 5d, 5e, 5f, 5h, 5i, 5j
- 3) Revised Supplementary Fig. 6f, 6g, 6h, 6i, 6j, 6k, 6l, 6m, 6n

Point-by-point response to the reviewer's comments

The responses are indicated in RED

Reviewer #2 (Remarks to the Author):

The authors addressed Reviewer 2's concerns about sex-specific modeling by performing karyotype and X/Y chromosome analyses and identifying Huh1 as a suitable male human liver cell line for comparison with the female HRO cells. Huh1 cells were added because the previously used Huh7 line lacked the Y chromosome and thus could not accurately represent a male model; in contrast, Huh1 cells possess a normal male karyotype, including functional KDM6C/UTY expression. This is a crucial limitation when using cell lines to study sex differences. The authors performed also a new RNA-seq and ChIP-seq experiments in Huh1 cells, confirming that KDM6A regulation in males differs from females and is largely independent of its demethylase activity. The mechanistic evidence was expanded by analyzing HNF4A–KDM6A interactions and CREBH binding, demonstrating a sex-specific regulatory axis in lipid and cholesterol metabolism. The title has been modified to clarify the statements about sex dimorphism, methodological data and data-availability details have now been added.

There are still some remaining inconsistencies or concerns.

1. Sex-dimorphism generalization remains overstated. Although the authors revised the title and toned down some claims, parts of the Discussion still generalize sex-specific conclusions. Many experiments proving “sex specificity” rely on only one female (HRO) and one male (Huh1) liver cell line, with Huh-7 remaining in some cases. Since all are cancer-derived and have abnormal karyotypes, it is not certain that the observed effects truly represent physiological male–female differences. The study still partially equates “cell line sex” with biological sex dimorphism, which remains an overextension. I thus suggest “female” and “male” cell line to be replaced by “cell line derived from a female or from a male patient “

Reply: We agree with the reviewer's point. We have changed the female and male cell line to 'cell line derived from a female or male patient' in the revised manuscript.

2. Remaining limited validation of HNF4A–KDM6A–CREBH axis in vivo. The mechanistic link between KDM6A, HNF4A, and CREBH is well supported in cell models, but less robust in mouse experiments. For instance, ChIP-seq and CUT&Tag evidence in mice support co-regulation of a few target genes (e.g., Apo1), but direct biochemical evidence of complex formation in vivo is not shown. This weakens the conclusion that the full “KDM6A–HNF4A–CREBH regulatory axis” is sex-specific and functional in the intact liver.

Reply: We agree with the reviewer that mechanistic evidence for the KDM6A–HNF4A–CREBH axis is more complete in human cell models than in mouse liver. In mice, direct KDM6A chromatin occupancy could not be assessed due to the lack of ChIP-grade antibodies suitable for mouse liver tissue, despite testing five commercial antibodies in our lab.

Nevertheless, multiple independent *in vivo* datasets support the same regulatory framework. Female LKO mice exhibit sex-specific alterations in lipoprotein profiles that align with pathway annotations derived from transcriptomic analyses in female-derived human liver cells. Transcription factor enrichment analyses in female LKO mouse liver consistently identified HNF4A and CREBH, mirroring results from HRO cells. Importantly, we identified and validated a ChIP-grade CREBH antibody functional in both mouse and human samples. CREBH ChIP-seq demonstrated genome-wide loss of CREBH binding upon KDM6A depletion in both human liver cell lines and female LKO mouse livers, providing direct *in vivo* support for CREBH regulation downstream of KDM6A.

We now explicitly acknowledge the absence of mouse KDM6A ChIP-seq and proteomic interaction data as a limitation in the Discussion.

3. Statistical and replication transparency. Although the Methods section now provides more statistical details, biological replicates and batch numbers for some omics experiments (especially ChIP-seq and CUT&Tag) remain unclear. The paper states that experiments were “repeated twice,” which may not be sufficient for reproducibility in multi-omics comparisons or should be commented.

Reply: We apologize for this unclear description in the methodology part. We have repeated the CREBH ChIP-seq in the HRO and mouse liver samples in the second revision. All the RNA-seq, ChIP-seq and CUT&TAG results in the current manuscript now contain at least 3 biological replicates (mostly n=4, all original data uploaded). The real-time PCR results in the manuscript were repeated at least twice and contain at least 3 biological replicates. We have adjusted the methodology description in the revised manuscript. The number of replicates in each figure is described in the figure legend to further enhance the clarity.

4. To overcome some of the limitations that might prove to be difficult experimentally, I suggest to include a final paragraph on study limitations that would strengthen the Discussion. It should explicitly acknowledge:

the restricted number of human models used

potential confounding due to cancer-derived chromosomal instability,

the need for validation in primary hepatocytes or organoid systems, and

that the mechanistic interactions among KDM6A, HNF4A, and CREBH remain partly correlative.

Reply: We have added ‘limitations of the study’ to the discussion part, which now includes all the issues raised by the reviewer 2.

Reviewer #3 (Remarks to the Author):

I would like to thank the authors for their efforts to improve the manuscript. Although they have included new and relevant data, I still have several points that need to be addressed.

In Figure 1J, the authors claim to show that the composition of VLDL is altered. However, Figure 1J presents plasma cholesterol levels during the VLDL secretion assay. Since the cholesterol levels at time point 0 are already much higher in the LKO mice compared to the WT, this conclusion cannot be made. It appears that the relative increase in plasma cholesterol during the poloxamer experiment is similar between the two groups.

To confirm that the VLDL composition is indeed altered, the authors should isolate the VLDL particles by centrifugation and determine the levels of apoB100, triglycerides (TG), total

cholesterol (TC), and phospholipids in these particles. Only then can they conclusively state that the VLDL composition is changed.

Reply: Thanks for the suggestions for reviewer 3. We agree that Fig. 1j alone does not directly demonstrate altered VLDL composition. To directly address this concern, we analyzed triglyceride and cholesterol levels within FPLC-isolated VLDL fractions from the same plasma samples. This enabled precise calculation of the VLDL cholesterol-to-triglyceride ratio independent of particle number.

Rebuttal Figure 1. VLDL cholesterol to triglyceride ratio during the VLDL production assay.

These analyses revealed a significantly increased VLDL cholesterol/triglyceride ratio in LKO mice throughout the VLDL production assay, demonstrating altered VLDL composition (**Rebuttal Fig. 1**, added to new **Supplementary Fig. 2i**). This dataset has now been incorporated into the revised manuscript.

In addition, what are the plasma ApoB100 levels in these mice? I expect an increase, which suggests more VLDL particles rather than a change in particle composition.

Rebuttal Figure 2. A) ELISA analysis of mouse plasma ApoB in the floxed and LKO mice after P-407 injection. B) Illustration of single-particle profiling technique to quantify plasma VLDL particles. C) Relative VLDL particle numbers between WT and LKO mice.

Reply: Thanks for the reviewer's suggestion. To assess whether altered plasma lipids reflected changes in VLDL particle number, we first measured ApoB100 using a C-terminal-specific ELISA as suggested by reviewer 3. ApoB100 levels were significantly higher in LKO mice. To be noted, ApoB100 levels in both WT and LKO mice were not increased throughout the VLDL production assay, uncoupled from both VLDL triglyceride and cholesterol results (**Rebuttal Fig. 2A**). In the rodents, both ApoB100 and ApoB48 can be structural component of VLDL, unlike humans which only have ApoB100. Therefore the ApoB100 alone does not capture total murine VLDL particles due to the substantial contribution of ApoB48-containing VLDL particles¹ (more than 50%).

We therefore applied single-particle profiling (SPP) (Taras, et al. Nature Biotechnology 2024)², a recently developed state-of-the-art biophysical method that directly quantifies VLDL particle numbers independent of ApoB isoform composition. Briefly, 100 ul plasma from each mouse was gradient ultra-centrifuged for 20 hours and VLDL fraction was collected, purified and fluorescently stained using a specific chemical called NR12S. By monitoring the continuous emission signal fluctuation from multiple fluorescent channels, the technique was able to directly quantify the VLDL particle numbers (Taras, et al. Nature Biotechnology 2024). The SPP analysis demonstrated comparable VLDL particle numbers between WT and LKO mice,

indicating that elevated VLDL cholesterol reflects altered particle composition rather than increased particle abundance (**Rebuttal Fig. 2B and 2C**, added to new **Fig. 1k**).

I disagree that LCAT reduction explains the reduction in plasma TG levels. LCAT deficiency increases plasma TG levels but attenuates diet-induced ApoB-containing lipoproteins. Reply: Thanks for the comments from reviewer 3. It was indeed reported that in the western diet fed mice, LCAT deficiency had reduced plasma triglyceride³. To be noted, the LCAT KO mice showed a sex specific phenotype in plasma triglyceride, the LCAT KO male mice had higher plasma triglyceride level comparing to WT as pointed out by reviewer 3, but the female LCAT KO mice showed no statistically different plasma triglyceride levels from the WT littermates⁴. The LCAT difference was observed in the female but not male LKO mice in our study.

We agree with the reviewer that the LCAT reduction does not solely explain the triglyceride level in our mice, it is rather the transcription network affected by KDM6A which defines the cholesterol and triglyceride phenotypes in combination.

What is the evidence that bile acid synthesis is reduced, not only in gene expression data? It suggests performing flux analysis of bile acids and determining bile acid concentration over time.

Rebuttal Figure 3. FPLC analysis of A) unconjugated and B) conjugated bile acids and C) C4 in the WT and LKO mice plasma.

Reply: Thanks for the comments from the reviewer. We agree that gene expression data alone cannot establish reduced bile acid synthesis. Accordingly, we did not claim ‘reduced bile acid synthesis’ per se but rather used ‘reduced expression of bile acid synthesis genes’. In our study, we could not fully address whether reduced bile acid synthesis genes contributed to the phenotype of the mice (further discussed in the revised manuscript).

The cholesterol phenotype in the liver and lipoprotein could be contributed by multiple mechanisms. To further address this issue, we analyzed the serum bile acid composition in the WT and LKO female mice using HPLC. The results showed that the elevated bile acids in the LKO female serum are mainly conjugated bile acids. All TCA, TDCA, TCDCA, TUDCA and the rodent-specific TMCA bile acids increased more than 10 times in the female LKO mice, while the unconjugated bile acids remained unchanged (**Rebuttal Fig. 3A-3B**). These data, along with the decreased bile acid excretion and transporter genes (*Ugt1a9*, *Abcc6*, *Sult2a3*, *Sult2a5*, *Sult2a8*, *Cyp2c37*, etc) strongly indicate defect of bile acid excretion in the liver,

leading to the accumulation and spillover of the conjugated bile acids into the blood. It is reasonable to hypothesize that the accumulated bile acids in turn repressed the further bile acid synthesis (i.e. via activating FXR to inhibit the key enzyme *Cyp7a1*) which further aggravated the cholesterol accumulation in the liver. To be noted, based on our data, CREBH binding at *Cyp7a1* locus was slightly reduced by KDM6A loss, suggesting a cistromically direct regulation of CREBH on *Cyp7a1* in the LKO liver.

Moreover, as a result of the cholesterol and bile acid accumulation, the transcriptomic results in the liver showed elevated inflammation and fibrosis genes (i.e. *Cd68*, *Cd14*, *Cxcl1*, *Ccl2*, *Saa2*, *Colla1*, *Colla2*, *Tgfb1*, *Tgfb2*, etc.). Taken together, the marked accumulation of conjugated bile acids, the reduced expression of bile acid excretion and transporter genes, and the induction of inflammatory and fibrotic genes in the liver are consistent with a cholestatic phenotype in female LKO mice. In such condition, the bile acid flux analysis will not be able to accurately measure the bile acid synthesis rate in the LKO mice. Taking a step back, even if we were able to use bile acid flux analysis to prove reduced bile acid synthesis, we still cannot conclude that such phenotype were the cause or consequence. Because the reduced bile acid synthesis could be an adaptive response to the hyper bile acid levels in the liver which trigger the negative regulatory loop (i.e. via FXR).

Despite that, we have made extra efforts using HPLC to measure a well established surrogate marker 7 α -hydroxy-4-cholesten-3-one (C4) to reflect the bile acid synthesis rate in the plasma samples of the WT and LKO mice⁵ (**Rebuttal Fig. 3C**). C4 is an intermediate product during bile acid synthesis catalysed by CYP7A1 from cholesterol⁵. Surprisingly, the C4 level is significantly higher in the LKO mice. However, this data does not necessarily mean the LKO liver bile acid synthesis rate is higher. One crucial enzyme to convert C4 to bile acid is CYP8B1, which is also reduced in the LKO mice. It is likely that the reduced rate-limiting enzyme of *Cyp8b1* may create a bottleneck to slow down the major branch of C4 utilization/clearance and lead to accumulated C4 in the liver and blood⁶, despite reduced C4 biosynthesis (hypothetically). Moreover, the injury of liver in cholestasis condition may lead to spillover of the cholesterol intermediates into the blood, causing the C4 elevation. Consistently, it was indeed reported that in cholestasis patients, the C4 correlation with bile acid levels seems to be uncoupled⁷ (i.e. high blood bile acid levels with ultra low C4 levels). We have added these data into the revised manuscript (**Supplementary Fig. 2p, 2q, 2r, 2s, 2t**) and discussed them further in the discussion.

We acknowledge the limitation of this study pointed out by the reviewer 3 that the hypothesis relies much on mRNA expression level changes. We have made efforts to improve this by validating some of the key enzymes using western blot and confirming the key phenotypes with multiple advanced techniques (HPLC, SPP, etc). We have also included a new session '**Limitation of the study**' in the discussion part of the revised manuscript to notify our readers about this major issue raised by the reviewer 3.

Reviewer #4 (Remarks to the Author):

Reply: We thank Reviewer 4 for the comments.

References:

1. Teusink, B. *et al.* Stimulation of the in Vivo Production of Very Low Density Lipoproteins by Apolipoprotein E Is Independent of the Presence of the Low Density Lipoprotein Receptor *. *J. Biol. Chem.* **276**, 40693–40697 (2001).
2. Sych, T. *et al.* High-throughput measurement of the content and properties of nano-sized bioparticles with single-particle profiler. *Nat. Biotechnol.* **42**, 587–590 (2024).
3. Karavia, E. A., Papachristou, D. J., Kotsikogianni, I., Triantafyllidou, I.-E. & Kypreos, K. E. Lecithin/cholesterol acyltransferase modulates diet-induced hepatic deposition of triglycerides in mice. *J. Nutr. Biochem.* **24**, 567–577 (2013).
4. Sakai, N. *et al.* Targeted Disruption of the Mouse Lecithin:Cholesterol Acyltransferase (LCAT) Gene: GENERATION OF A NEW ANIMAL MODEL FOR HUMAN LCAT DEFICIENCY*. *J. Biol. Chem.* **272**, 7506–7510 (1997).
5. Gälman, C., Arvidsson, I., Angelin, B. & Rudling, M. Monitoring hepatic cholesterol 7 α -hydroxylase activity by assay of the stable bile acid intermediate 7 α -hydroxy-4-cholesten-3-one in peripheral blood. *J. Lipid Res.* **44**, 859–866 (2003).
6. Li, T. & Apte, U. Bile acid metabolism and signaling in cholestasis, inflammation and cancer. *Adv. Pharmacol. San Diego Calif* **74**, 263–302 (2015).
7. Braadland, P. R. *et al.* Suppression of bile acid synthesis as a tipping point in the disease course of primary sclerosing cholangitis. *JHEP Rep.* **4**, 100561 (2022).